# Immunoproteasome expression is associated with better prognosis and response to checkpoint therapies in melanoma

Shelly Kalaora [1,9], Joo Sang Lee[2,3,9], Eilon Barnea[4], Ronen Levy[1], Polina Greenberg[1], Michal Alon[1], Gal Yagel [1], Gitit Bar Eli[1], Roni Oren[5], Aviyah Peri[1], Sushant Patkar[2], Lital Bitton[1], Steven A. Rosenberg[6], Michal Lotem[7], Yishai Levin[8], Arie Admon[4], Eytan Ruppin [2✉] & Yardena Samuels[1✉]

Predicting the outcome of immunotherapy treatment in melanoma patients is challenging. Alterations in genes involved in antigen presentation and the interferon gamma (IFNγ) pathway play an important role in the immune response to tumors. We describe here that the overexpression of *PSMB8* and *PSMB9*, two major components of the immunoproteasome, is predictive of better survival and improved response to immune-checkpoint inhibitors of melanoma patients. We study the mechanism underlying this connection by analyzing the antigenic peptide repertoire of cells that overexpress these subunits using HLA peptidomics. We find a higher response of patient-matched tumor infiltrating lymphocytes against antigens diferentially presented after immunoproteasome overexpression. Importantly, we find that *PSMB8* and *PSMB9* expression levels are much stronger predictors of melanoma patients' immune response to checkpoint inhibitors than the tumors' mutational burden. These results suggest that *PSMB8* and *PSMB9* expression levels can serve as important biomarkers for stratifying melanoma patients for immune-checkpoint treatment.

[1] Department of Molecular Cell Biology, Weizmann Institute of Science, Rehovot, Israel. [2] Cancer Data Science Lab, National Cancer Institute, Bethesda, MD, USA. [3] Samsung Medical Center, Sungkyunkwan University School of Medicine, Suwon, Republic of Korea. [4] Department of Biology, Technion, Haifa, Israel. [5] Department of Veterinary Resources, Weizmann Institute of Science, Rehovot, Israel. [6] The Surgery Branch, National Cancer Institute, Bethesda, MD, USA. [7] Sharett Institute of Oncology, Hadassah Medical School, Jerusalem, Israel. [8] Nancy and Stephen Grand Israel National Center for Personalized Medicine, Weizmann Institute of Science, Rehovot, Israel. [9] These authors contributed equally: Shelly Kalaora, Joo Sang Lee. ✉email: eytan.ruppin@nih.gov; Yardena.Samuels@weizmann.ac.il

A high neo-antigen load in tumors has been associated with enhanced response to immunotherapy[1]. However, tumors containing equally high mutational loads exhibit variable immune responses, suggesting that additional factors determine the development of T-cell reactivity[2]. Indeed, abnormalities in antigen presentation and the interferon gamma (IFNγ) pathways, for example, have been shown to affect the response of melanoma patients to immunotherapy[2–6].

To identify additional elements that influence the response to immunotherapy, we focused on the immunoproteasome, which is specifically documented to degrade cell proteins to generate peptides for antigen presentation[7]. PSMB8 and PSMB9 are two of the immunoproteasome (IP) subunits. Under inflammatory conditions (for example, in the presence of cytokines such as IFNγ and TNFα), they replace the constitutively expressed proteasome subunits PSMB5 and PSMB6[8]. Since the catalytic activities of the two immunoproteasome subunits differ from those of PSMB5 and PSMB6, the cleavage specificity of the proteasome is altered during inflammation, the result of which is the creation of different peptides[9]. Numerous studies have evaluated whether known immunogenic tumor-associated antigen (TAAs) are processed by the regular proteasome or the immunoproteasome. Some analyses indicate that immunogenic antigens are created only by the immunoproteasome[10–13], but others suggest that there are immunogenic antigens that are processed by both the immunoproteasome and the regular proteasome[12, 14] or exclusively by the regular proteasome[12–14]. Some immunogenic antigens require only some of the immunoproteasome subunits for their processing[15] or exclusively require only particular subunits to be present[16] (also called the intermediate proteasome). Thus, immunogenic antigens are produced by different forms of the proteasome, but it is not known which antigens are responsible for tumor rejection in the context of the tumor.

A previous study has demonstrated that the immunoproteasome subunits PSMB8 and PSMB9 are overexpressed in melanoma cell lines[13]. Here we studied their genomic and transcriptomic alteration in melanoma patients analyzing The Cancer Genome Atlas (TCGA) data and observed a high frequency of amplification and overexpression of these genes. Tripathi et al.[17] had reported that the reduced expression of immunoproteasome subunits in non-small cell lung carcinoma is associated with poor outcome. We show here, for the first time, that the overexpression of these subunits is correlated with improved survival and better response to immune-checkpoint inhibitors in melanoma.

We hypothesized that the overexpression of immunoproteasome subunits may influence the production of HLA peptides, and that the new peptide repertoire may prompt a higher immune response. To test this hypothesis, we utilized HLA peptidomics to analyze the changes in the HLA peptide repertoire of melanoma cells due to PSMB8 and PSMB9 overexpression and determined the effects of these changes on the reactivity of patient infiltrating tumor lymphocytes (TILs). We found that when PSMB8 and PSMB9 are overexpressed, the repertoire of antigens presented is altered and that the immune response to the presented neo-antigens and TAAs that are differentially presented when PSMB8 and PSMB9 are overexpressed is higher.

## Results

### Overexpression of immunoproteasome subunits is correlated with improved melanoma patients' survival independent of mutational load, IFNγ, or T-cell infiltration. To assess the relationship between the expression levels of immunoproteasome subunits PSMB8 and PSMB9 and melanoma patient survival, we analyzed data from TCGA of 472 melanoma patients for whom

RNA-seq data and patient outcome were available (Supplementary Table 1). Our assessment of PSMB8 and PSMB9 mRNA expression levels in TCGA samples compared to GTEX healthy controls (Supplementary Table 2) revealed the overexpression of the two immunoproteasome (IP) subunits in the TCGA (t-test $P < 4.1E-57$ and $P < 7.3E-79$, respectively). We found a high correlation in PSMB8 and PSMB9 expression levels (Spearman $R = 0.90$, $P < 1E-15$) and a statistically significant co-occurrence of their overexpression (hypergeometric $P < 4.9E-55$, see Methods). This observation led us to define the immunoproteasome subunit expression levels as the summed expression of PSMB8 and PSMB9.

Our analysis shows that PSMB8 and PSMB9 expression is associated with better overall patient survival (Fig. 1a, logrank $P < 0.005$; the signal is robust with different subgrouping of patients, as shown in Supplementary Fig. 1). In contrast, tumor mutational load does not significantly correlate with patient survival (Fig. 1b, logrank $P < 0.27$). The association between IP subunit expression and patient survival remained significant even after controlling for multiple confounders, including patient age, race, sex, and tumor purity[18,19], while the expression of constitutive proteasome subunits (PSMB5 and PSMB6) showed no significant association (Supplementary Fig. 2, Supplementary Table 3).

We then evaluated, using CIBERSORT[20], whether the overexpression of IP subunits correlates with the abundance of various tumor-associated leukocyte subsets. PSMB8 and PSMB9 overexpression was found to highly associate with CD4+ and CD8+ T-cell infiltration, regulatory T-cells, NK cells and M1-macrophages (Fig. 1c), in agreement with a role for IP subunit overexpression in enhancing the immune response in the tumor. This association is maintained even when tumor purity is controlled for in a linear model (Supplementary Table 4). In addition, we observed a significant association between cytolytic activity (CYT score[4]) and IP subunit expression (Fig. 1c, Supplementary Table 4), but not for their constitutive counterparts, suggesting that the longer overall survival may indeed be associated with a stronger contribution of the immunoproteasome subunits to T-cell cytotoxicity.

Immunoproteasome expression is known to be closely associated with IFNγ or T-cell infiltration, but it remains unclear whether the IP subunits independently contribute to patient survival. As expected, we observed that IFNγ signature, expression of T-cell-related genes and CD8+ T-cell infiltration (as determined by CIBERSORT) all also show a significant association with patient survival (Supplementary Fig. 3). However, these latter associations vanish when tumor purity is controlled for in the Cox model (Table 1), while the association found for the IP subunits remains. Moreover, a multivariate Cox model of IP subunit expression together with IFNγ and T-cell infiltration shows a significant association of IP expression with patient survival, but not for IFNγ or T-cell infiltration. These results testify that IP subunit overexpression in cancer cells is independent of IFNγ or T-cell infiltration (Supplementary Figs. 4, 5, see Supplementary Note 1) and is a strong independent prognostic biomarker for melanoma patient survival.

### The effect of immunoproteasome subunit overexpression on the immune response of autologous TILs. To test our hypothesis that overexpression of IP subunits derives alternative, more immunogenic peptides, we overexpressed both PSMB8 and PSMB9 (OE) or a vector control (EV) in three different melanoma cell lines (108T, 12T and A375) (Supplementary Fig. 6a–c). In a complementary experiment, we increased the expression of the endogenous immunoproteasome subunits by treating 108T

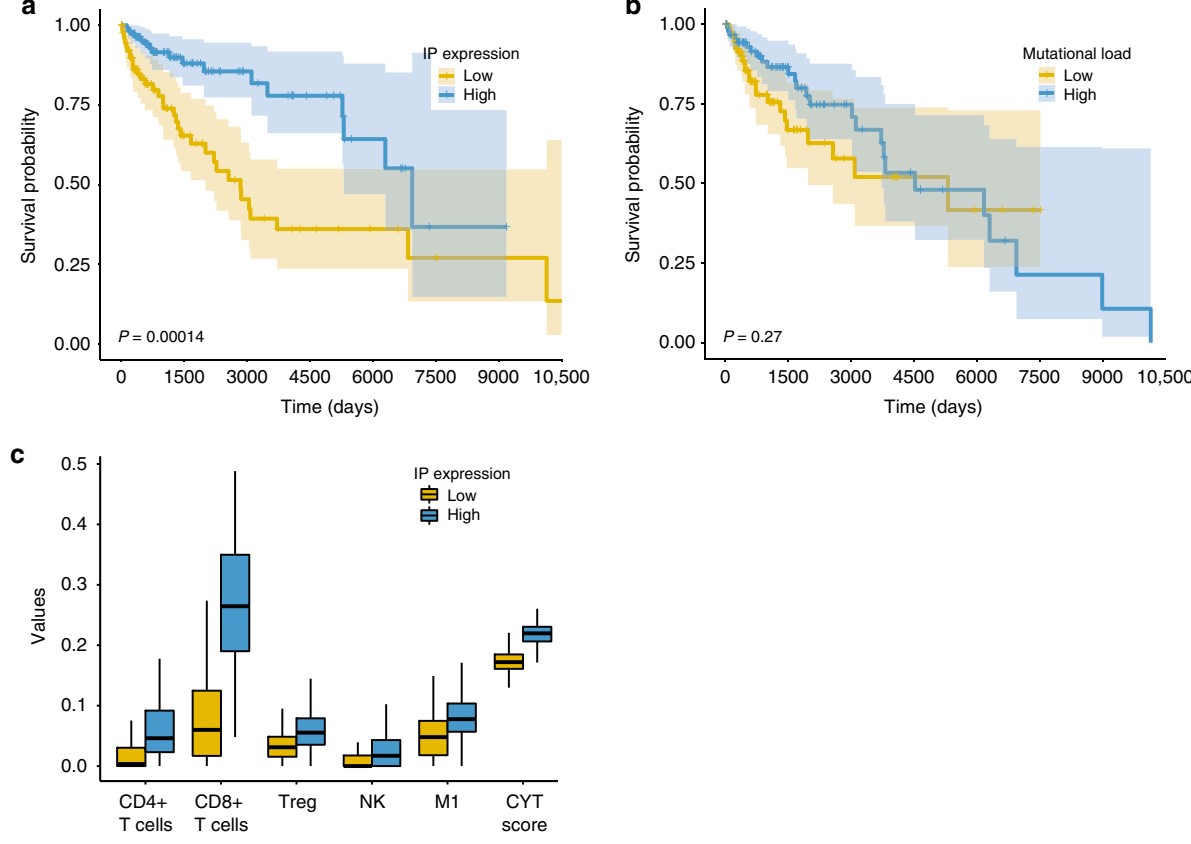

**Fig. 1 IP subunits expression is associated with better prognosis. a**, **b** Kaplan–Meier plot of TCGA melanoma patients ($n = 472$), where the survival of patients with high (**a**) IP expression, and (**b**) Mutational load (top tertile; blue) is compared with that of the patients with the low counterparts (bottom tertile; yellow) (logrank $P = 0.00014$ and $P = 0.27$ with median survival time difference 4094 and 783 days, respectively). **c** The estimated abundance of CD4 + T-cell, CD8+ T-cell, regulatory T-cell (Treg), NK cell (NK), M1 macrophage (M1), and cytolytic score (CYT score) of patients with high IP subunits expression (top tertile; blue) are higher compared to patients with low IP subunits expression (bottom tertile; yellow) (FDR-corrected Wilcoxon ranksum $P < 0.05$).

**Table 1 Comparative Cox regression analysis of IFNγ, T-cell infiltration, and IP expression in TCGA melanoma patients.**

|  | Univariate model: HR | Univariate model: P | Univariate model + purity: HR | Univariate model + purity: P | Multivariate model: HR | Multivariate model: P |
|---|---|---|---|---|---|---|
| IFNγ | 0.709 | 0.000964 | 0.763 | 0.094754 | 1.042 | 0.861773 |
| CD8+ T-cell | 0.766 | 0.015118 | 0.877 | 0.319634 | 1.097 | 0.576941 |
| CD8A | 0.694 | 0.000629 | 0.706 | 0.063550 | 0.886 | 0.628285 |
| CD4 | 0.682 | 0.000536 | 0.678 | 0.078271 | 0.721 | 0.122652 |
| CD3G | 0.695 | 0.000953 | 0.695 | 0.114972 | 0.871 | 0.627104 |
| Mutational load | 0.796 | 0.074994 | 0.806 | 0.089139 | 0.842 | 0.194167 |
| *PSMB8* | 0.683 | 0.001128 | 0.753 | 0.036950 | – | – |
| *PSMB9* | 0.649 | 8.59E-05 | 0.651 | 0.009516 | – | – |
| *PSMB8/9* | 0.646 | 0.000138 | 0.677 | 0.011117 | 0.646 | 0.045741 |

The table shows the hazard ratio (HR) and corresponding Wald *p*-values (P) of the three Cox models that we considered, namely: Univariate model (each variable as single factor), Univariate model + purity (each variable and purity as additional factor) and multivariate models (all variables all together) with IFNγ, computationally estimated CD8+ T-cell infiltration, T-cell related gene expression (CD8A, CD4, and CD3G), *PSMB8*, and *PSMB9* expression, and the summed expression of *PSMB8/9* as independent variables for explaining patient survival.

and 12T cells with IFNγ (Supplementary Fig. 6d). To ensure the immunoproteasome is active in the cells overexpressing the IP subunits, we used fluorescent peptides that can be cleaved by the chymotrypsin-like activity of PSMB5 and PSMB8 (Suc-LLVY-AMC) and a substrate that is specifically cleaved by PSMB9 (Suc-PAL-AMC) (Supplementary Fig. 7). In all three tested cell lines, we observed increased cleavage, represented by relative fluorescence units (RFUs), in the cells overexpressing the immunoproteasome subunits compared to the empty control. This finding indicates that the overexpressed immunoproteasome subunits

were incorporated into the proteasome complex and were active. The change in RFU is correlated to the level of change in immunoproteasome expression, as the greatest change was noted in 12T, then A375, and last 108T, which paralleled the decreasing trajectory noted in their endogenous immunoproteasome expression in the parental cells (EV). As it is known that IFNγ induces the expression not only of the immunoproteasome but also of the HLA complex[21,22], we assessed, by flow cytometry, the HLA expression levels in both IFNγ-treated cells and cells overexpressing the immunoproteasome (Supplementary Fig. 8). As

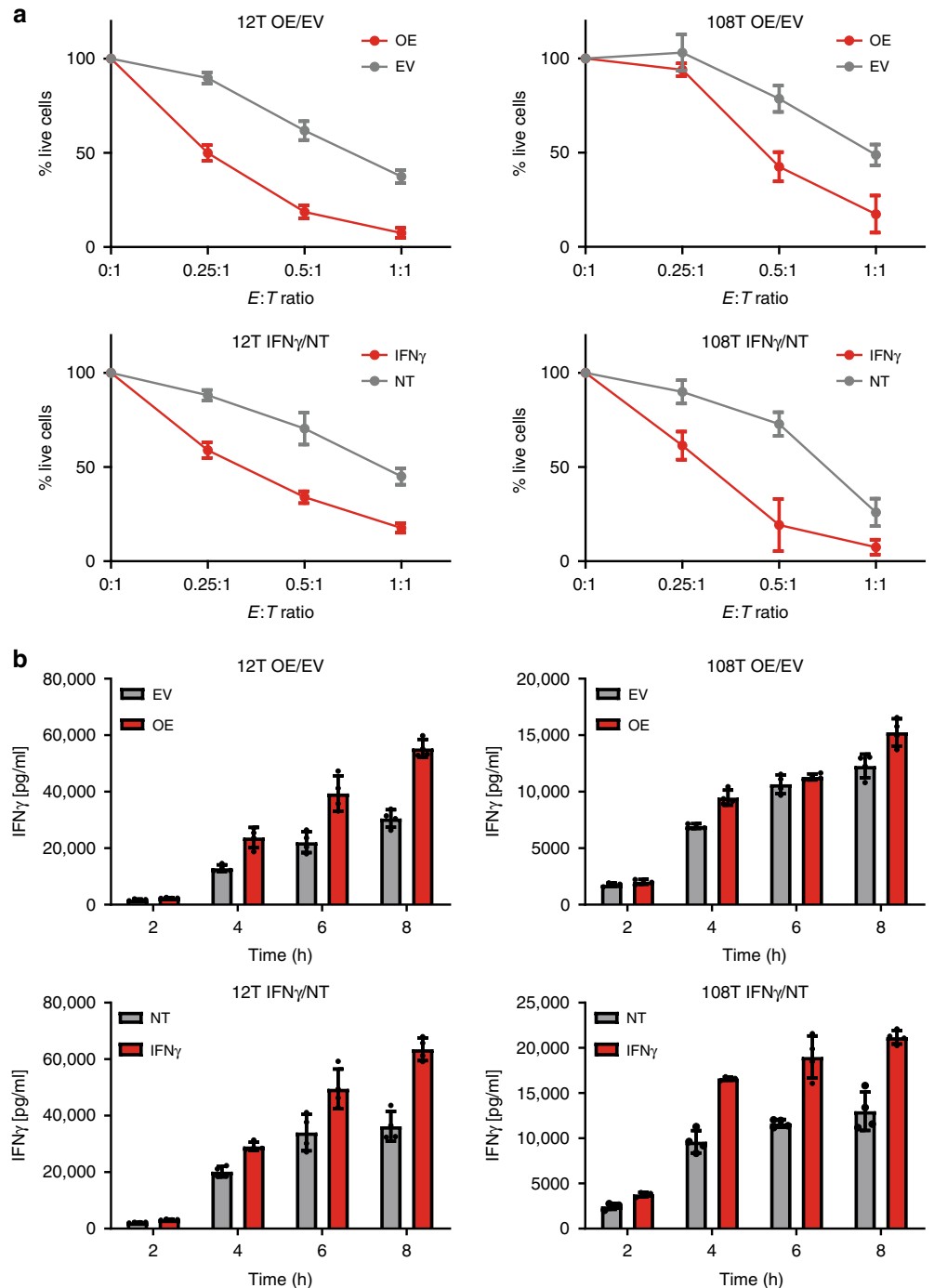

**Fig. 2 Reactivity toward cells with overexpression of immunoproteasome subunits is higher compared to control. a** Cells with immunoproteasome overexpression compared to empty vector control or cells treated with IFNγ compared to untreated cells were co-cultured in different ratios with autologous TILs. (E:T, different effector to target ratios). Number of live cells were counted after 8 or 12 h for 12T and 108T, respectively. **b** IFNγ secretion was measured after 2, 4, 6, and 8 h of co-culture of cells with immunoproteasome overexpression compared to empty vector control or cells treated with IFNγ compared to untreated cells. Data were analyzed from n = 3 biological repeats per each condition and represented as mean ± SD.

expected, HLA levels rose after IFNγ treatment but no change was observed due to IP overexpression.

To assess the effect of overexpressing the immunoproteasome subunits or their induction by IFNγ treatment on the immune response, we compared the ability of the autologous TILs to lyse cells either overexpressing the IP subunits or treated with IFNγ. To this end, we co-cultured the cells with increasing concentrations of TILs and counted the live cells that remained. As seen in Fig. 2a, autologous TILs killed the melanoma cells overexpressing

the IP subunits more efficiently than the vector control. Likewise, TILs were more competent at killing cells treated with IFNγ compared to untreated cells. Similar results were observed for IFNγ secretion measurements (Fig. 2b). These observations support our hypothesis that immunoproteasome overexpression produces a better immune response.

**Identification of HLA class I antigens presented in the presence of active immunoproteasome subunits.** The degradation of

cellular proteins by the proteasome and immunoproteasome is central to the generation of HLA-associated peptides for presentation to T-cells[8]. Hence, the constitutive proteasome and the immunoproteasome subunits may each promote the presentation of a distinct peptide repertoire and, consequently, a different immune response outcome. Our observation of increased killing of cells overexpressing the IP subunits led us to hypothesize that these cells may produce a different, possibly more immunogenic antigen repertoire. We thus investigated the role that overexpression of immunoproteasome subunits plays in melanoma cell neo-antigen and TAA presentation and immune response elicitation.

To test this, we employed HLA peptidomics to profile HLA-I-bound antigens in the three cellular systems noted above, as previously described[23–25]. In total, we identified 17,501 unique HLA peptides: 8418 peptides in 108T cells, 6205 peptides in 12T cells and 5575 peptides in A375 cells. These peptides were derived from 8968 different proteins: 5780 proteins in 108T cells, 3910 in 12T cells and 2362 in A375 cells (Supplementary Tables 5–10). Clustering the peptides into six different clusters showed, as expected, reduced amino acid complexity at the second and ninth anchor residues, which match the binding motifs of the cells' HLA alleles (Supplementary Fig. 9–11). In all samples, more than 98% of the peptides were clustered using the Gibbs clustering analysis, with 83–95% of the peptides predicted to bind the patients' HLA alleles using NetMHCpan, indicating that those peptides are actual HLA ligands (Supplementary Table 5). Peptides that did not cluster using the Gibbs clustering analysis and were not predicted to bind the HLA by NetMHCpan were excluded from all further analyses. The length distribution of the identified peptides matched the expected distribution for class I peptides (Supplementary Fig. 12). As anticipated, we observed a greater similarity between samples of the same experiment than between samples of the same cell line (Supplementary Fig. 13). Of all the identified HLA-bound peptides, two neo-antigens were identified in 12T cells, with 123, 117, and 59 TAAs identified in 108T, 12T, and A375 cells, respectively. These TAAs were derived from 49 108T proteins, 45 12T proteins, and 32 A375 proteins, and derived from known cancer/testis and previously described melanoma antigens[26–28]. Both neo-antigens derived from missense mutations: DANSFLQSV from a P677S mutation in the mediator complex subunit 15 (MED15) gene, and KLFEDRVGTIK from a S123L mutation in the TPD52 like 2 (TPD52L2) gene[23,25]. We validated the identification of the neo-antigens by comparing their MS/MS spectra with that of synthetic peptides (Supplementary Fig. 14) and by spiking a stable isotopically labeled peptide that co-eluted with them (Supplementary Fig. 15).

To explore the changes in the peptide repertoire of cells overexpressing the immunoproteasome subunits (OE) compared to the control (EV), we searched the peptides that exhibited a change in their intensity between groups. We first looked at all the peptides and then focused on those whose amount significantly changed (Two side student's t-test, permutation based FDR = 0.05, S0 = 1) (Fig. 3). We observed more presented peptides in the cells overexpressing IP compared to the vector control (3164 vs. 1358 in 12T, 4023 vs. 1256 in 108T, 3015 vs. 2559 in A375, respectively). Similar results were obtained for the cohort of peptides that significantly differentially presented (1094 vs. 129 in 12T, 623 vs. 114 in 108T, 340 vs. 184 in A375, respectively). We noted more neo-antigens and TAAs after overexpression of the immunoproteasome subunits in 12T and 108T, and a similar number in A375 cells. From the total TAA pool: 68 vs. 28 peptides in 12T, 67 vs. 14 peptides in 108T and 50 vs. 51 peptides in A375, reflecting after vs. before results. From the significantly differentially presented TAAs: 25 vs. 2 peptides

in 12T, 2 vs. 3 peptides in 108T, 9 vs. 2 peptides in A375, reflecting after vs. before results.

As expected, peptides differentially presented by the IP overexpressing cells had a clear tendency to cluster to alleles with chymotryptic-like motifs in their C-terminus (A/F/I/L/M/V/Y amino acids in their c-terminus), and less to alleles with tryptic-like motifs (K/R amino acids in their C-terminus), except in the case of the 108T cells (in which most peptides clustered under the A*11:01 allele). As this is a known IP motif, this data substantiates our above biochemical analysis. In the 12T cells, more peptides matched the B*51:01 and C*01:02 alleles than the A*03:01 allele. In A375 cells, all alleles have chymotryptic-like motif in their C-terminus, but still more peptides had a Y/F/W (B*44:03 and B*57:01) at the end and less had a V/L (A*02:01), probably because the former alleles contain an aromatic side chain, which is more preferable for a chymotryptic-like cleavage. Generally, the alleles of cells in which the IP is overexpressed are more uniformly represented by peptides, creating a larger diversity of the presented alleles (Supplementary Fig. 16).

We performed a similar analysis on cells treated with IFNγ vs. non-treated cells (NT). Compared to the overexpression of the immunoproteasome subunits, treatment of cells with IFNγ, in order to induce IP expression, is accompanied by additional changes such as the induction of HLA molecules expression[21,22] (Supplementary Fig. 8) and changes in gene expression[29,30], making it more difficult to differentiate how IFNγ-induced IP expression affects the resulting peptide repertoire. Despite this challenge, we were able to show that, similar to the results seen for cells overexpressing the IP subunits, IFNγ-treated cells exhibit changes in the HLA peptide repertoire. Specifically, in 12T cells, more peptides were presented after treatment with IFNγ: 2200 compared to 1500 peptides in total, 938 compared to 486 in the significantly changed presentation cohort. In 108T, more peptides were presented without treatment with IFNγ: 3596 compared to 2916 peptides in total, 2144 compared to 1973 in the significantly changed presentation cohort. We observed more neo-antigens and TAAs after overexpression of the immunoproteasome subunits in 12T cells but not in 108T cells. From the total identified TAAs, 31 compared to 15 peptides in 12T and 46 compared to 51 peptides in 108T. From the significantly changed TAAs, 13 compared to 5 in 12T and 35 compared to 42 in 108T (Fig. 3).

We further observed the pronounced appearance of more peptides that match to the HLA-B alleles after treatment with IFNγ, in line with IFNγ's known preference to increase HLA-B expression over that of other HLA-I alleles. Interestingly, we identified in 12T a new cluster for the B*08:01 allele, which did not have a sufficient number of peptides to produce a cluster prior to IFNγ-treatment, as well as an increase in the representation of peptides that match the B*51:01 and C*01:02 alleles, reminiscent of the data obtained in cells overexpressing the IP subunits. We likewise observed in the IFNγ-treated 108T a new cluster for the HLA-B*55:01 allele, which did have sufficient peptides to produce a cluster before, and an increase in the representation of peptides that match the C*03:03 and C*07:02 alleles. Similarly to the change after overexpression of immunoproteasome subunits, the alleles are represented more uniformly by peptides after treatment with IFNγ, increasing the diversity of the presented alleles (Supplementary Fig. 16).

**Characterization of the immunogenicity of neo-antigens and TAAs presented by immunoproteasome-overexpressing cells.** To assess which of the identified HLA-bound TAAs and neo-antigens are immunoreactive, we tested the reactivity of autologous TILs to these peptides by pulsing synthetic equivalents

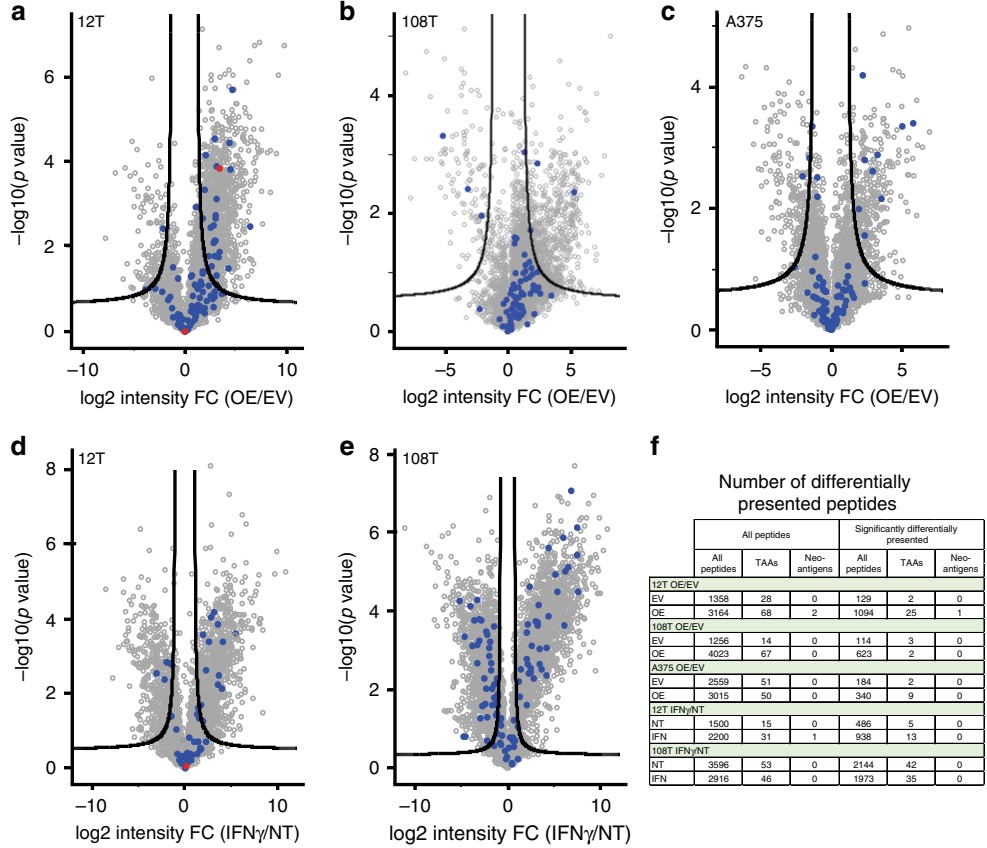

**Fig. 3 Differently presented peptide repertoire in cells with immunoproteasome overexpression.** Volcano plots were plotted to identify the peptides that were differentially presented by cells with overexpression of the immunoproteasome subunits (OE) compared to the control (EV) (**a–c**), or of the cells that were treated with IFNγ (IFNγ) compared to non-treated cells (NT) (**d**, **e**). Each HLA peptidomics experiment was done on three independent cell cultures. Peptides were determined as significantly changed if they passed statistical analysis (Two side student's t-test, permutation based FDR = 0.05, S0 = 1) and were found in the plot above the lines. TAAs were marked in blue dots and neo-antigens in red dots. The number of differentially presented peptides is indicated in the table (**f**).

onto EBV-transformed B-cells expressing matched HLA alleles and then co-culturing the B-cells with the autologous TILs. For both 12T and 108T, we detected a higher total reactivity of the TAAs and neo-antigens that were differentially presented by cells that overexpress the IP subunits compared to TAAs that were differentially presented by the control cells (Fig. 4 and Supplementary Fig. 17). This improved immune response is a result of the production of more immunogenic antigens by the cells with higher immunoproteasome expression. In 12T, we observed a 7.3-fold increase in total reactivity of the peptides differentially presented in cells with immunoproteasome over-expression (56,750 pg/ml vs. 7920 pg/ml in the EV control). This fold difference remains when we focus on the significantly differentially presented peptides. Similar results were obtained when we compared the peptides presented by IFNγ-treated cells to non-treated cells—a 94.8-fold increase in reactivity (32,643 pg/ml vs. 344 pg/ml) for all peptides and a 4.2-fold increase in the case of significantly differentially presented peptides (Fig. 4). In 108T, we observed an 8.9-fold increase in total reactivity between IP overexpressing cells compared to EV control cells and a 2.3-fold change between treated and non-treated cells for all peptides. A 2.4-fold change was found for the significantly differentially presented peptides between treated and non-treated cells. In the IP overexpressing cells compared to EV, there were not many differentially presented TAAs in 108T, and they were all non-reactive (Supplementary Fig. 17).

**Increased expression of the immunoproteasome subunits correlates with a better response to immune-checkpoint inhibitors.** We analyzed expression data from recent studies in which melanoma patients were treated with immune-checkpoint therapy (ICT)[2,31] to determine whether melanoma patient response to ICT is correlated with *PSMB8* and *PSMB9* levels. Analysis of an anti-CTLA4 (ipilimumab) cohort[31] showed a high correlation between *PSMB8* and *PSMB9* expression, as observed in the TCGA cohort (Spearman $R = 0.88$, $P < 1E-15$). Of the 35 samples for which gene expression, mutation and response annotation were available, we considered 26 samples (9 responders and 17 non-responders) for which tumor purity was >45%, to mitigate potential confounding effects from surrounding stromal or immune cells[19]. Importantly, we were able to confirm that the outcome of our analysis did not depend on purity thresholds (Supplementary Table 11). Anti-CTLA4 treatment showed a more durable benefit for patients that expressed high levels of *PSMB8* and *PSMB9* than for those with low levels ($P = 0.006$, Wilcoxon rank sum test). In comparison to other single genes, *PSMB9* was within the top 1.5% ($P < 0.01$, Wilcoxon rank sum test) and *PSMB8* within the top 4.4% ($P < 0.04$, Wilcoxon rank sum test) of genes that are significantly associated with durable clinical benefit after treatment with ipilimumab (Supplementary Table 12). In difference, the expression of the constitutive proteasome subunit *PSMB5* and *PSMB6* genes was not predictive of response to anti-CTLA4 treatment ($P = 0.5$, Wilcoxon rank sum test) (Fig. 5a).

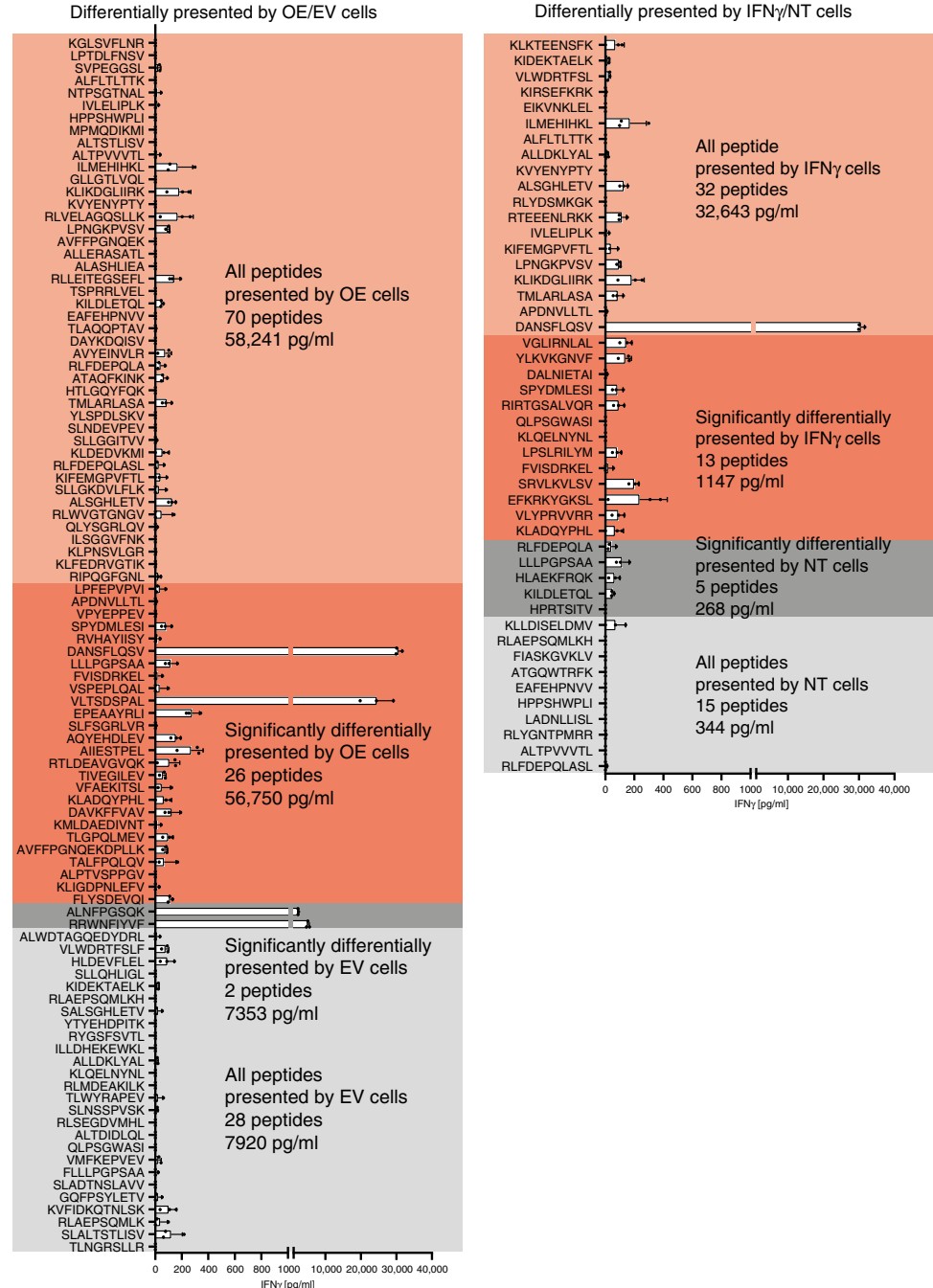

**Fig. 4 HLA peptide repertoire of 12T cells with overexpression of immunoproteasome subunits is more immunogenic compared to the control peptide repertoire.** All peptides whose fold change in intensity was greater or smaller than one by cells with immunoproteasome overexpression compared to empty vector control (**a**) or cells treated with IFNγ compared to untreated cells (**b**) were tested for their ability to elicit an immune response by the autologous TILs. The reactivity of each peptide was measured from $n = 3$ biological repeats per each peptide and represented as mean ± SD. The sum reactivity of each group of peptides, and the number of peptides in each group is indicated. Red and gray areas represent all peptides whose fold change intensity was greater or smaller than one by the OE/IFNγ cells and EV/NT cells, respectively (including the significantly changed peptides). Dark red and dark gray areas represent the peptides that were significantly differentially presented by the OE/IFNγ cells and EV/NT cells respectively. For each condition the sum reactivity was calculated for all peptides and for the ones that were significantly differentially presented.

We compared the predictive power of immunoproteasome subunit expression with known correlates of ICT response such as mutational load, IFNγ, and T-cell infiltration. The mutational load was associated with an improved outcome after anti-CTLA4 treatment (Fig. 5a), but less so than was the expression of immunoproteasome subunits (mutational load $P = 0.03$, Wilcoxon rank sum test). Moreover, we confirmed that the

expression levels of immunoproteasome subunit correlate with ICT response also when mutational load, tumor purity, IFNγ, and CD8+ T-cell abundance are controlled for via a partial correlation analysis (Kendall tau = 0.26, $P < 0.09$), pointing to the independent contribution of IP subunits to ICT response. Indeed, immunoproteasome subunit expression has superior predictive power (Area under the Curve (AUC) of Receiver

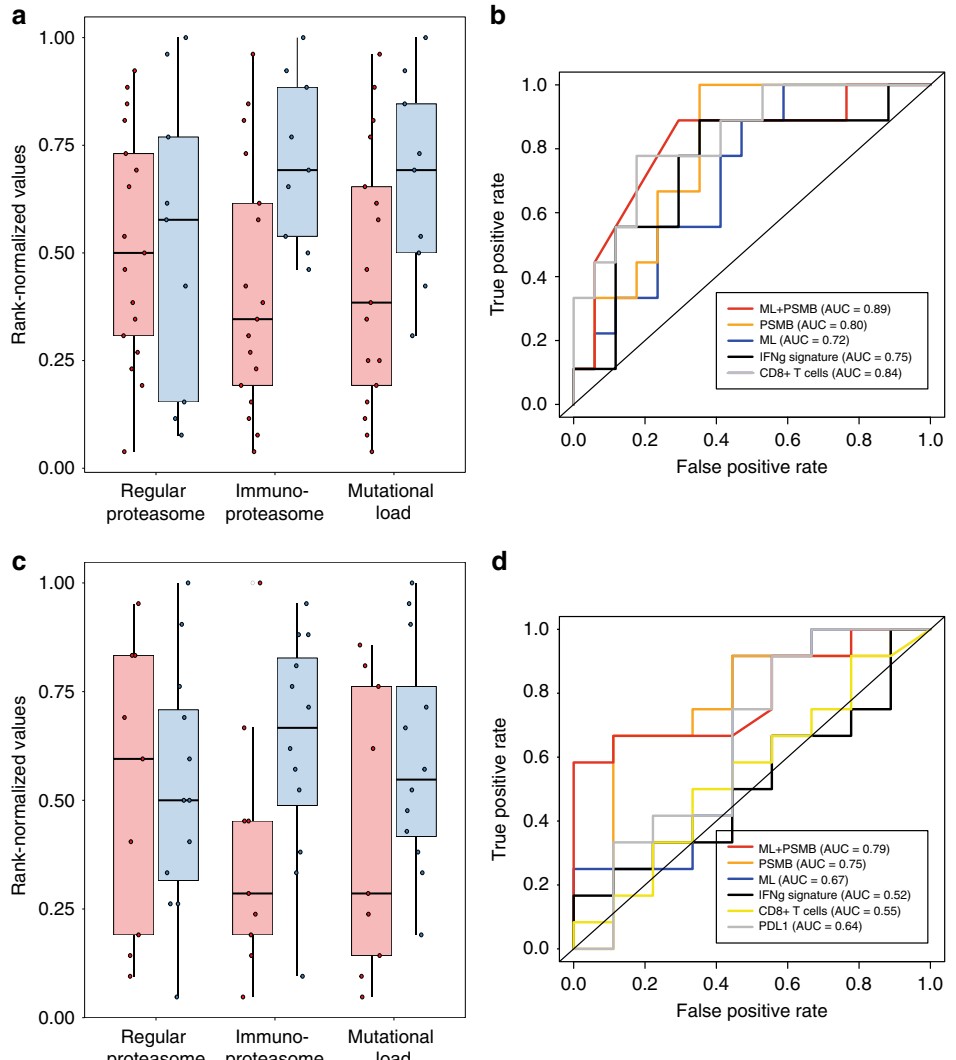

**Fig. 5 IP expression is associated with better response to anti-CTLA4 and anti-PD1 therapy. a** The responders to anti-CTLA4 therapy show significantly higher expression of IP subunits (one-sided Wilcoxon ranksum $P < 0.006$) and mutational load (one-sided Wilcoxon ranksum $P < 0.03$). No significant difference was observed in regular proteasome subunit expression (*PSMB5/6*; Wilcoxon ranksum $P > 0.5$). **b** The IP subunits expression is predictive of the response to anti-CTLA4 therapy, quantified by area under the curve (AUC = 0.80, orange) of receiver operating curve (ROC), superior or comparable to the mutational load (AUC = 0.72, blue) and IFNγ signature (AUC = 0.75, black). In combination with ML, IP (PSMB) is the most predictive (AUC = 0.89, red) followed by CD8T (AUC = 0.87) and IFNγ (AUC = 0.84, black) or CD8T as a single variable (AUC = 0.82, gray). Precision-recall curve is shown in Supplementary Fig. 18a. **c** The responders to anti-PD1 therapy show significantly higher expression of IP subunits (one-sided Wilcoxon ranksum $P < 0.03$). No significant difference was observed in regular proteasome subunit expression (*PSMB5/6*; one-sided Wilcoxon ranksum $P > 0.5$) and mutational load (one-sided Wilcoxon ranksum $P > 0.1$). **d** The IP (orange) is predictive of the response to anti-PD1 therapy (quantified by area under the curve (AUC = 0.75 of receiver operating curve (ROC)), superior to the mutational load (AUC = 0.67, blue), IFNγ signature (AUC = 0.52, black), CD8+ T-cell abundance (AUC = 0.55, yellow), and PDL1 expression (AUC = 0.64, gray). The combination of IP subunit expression and mutational load provides AUC = 0.79 (red). Precision-recall curve is shown in Supplementary Fig. 18b.

Operating Curve (ROC = 0.80)) compared to mutational load and IFNγ signature. Further, the combination of IP subunit expression with mutational load shows the highest prediction accuracy as compared to all other combinations with mutational load (Fig. 5b). This relationship remained robust when we considered all 35 samples without the purity cut-off (Supplementary Table 11).

The majority of the tumors (6/9) with high immunoproteasome subunit expression and high mutational load (>50th percentile) were found in patients who exhibited a durable clinical benefit following anti-CTLA4 therapy. Notably, none of the patients (0/8) whose tumors exhibited low immunoproteasome subunit expression and low mutational load

(<50th percentile) benefited from ipilimumab. Among the high-purity tumors (purity > 45%), the majority of the tumors (4/5) with high immunoproteasome subunit expression and high mutational load (>50th percentile) were from patients who had a durable clinical benefit, but none of the patients (0/5) with tumors with a low immunoproteasome subunit expression and low mutational load (<50th percentile) benefited from ipilimumab.

Next, we studied a cohort of patients treated with anti-PD1 antibodies pembrolizumab and nivolumab[2]. Here, too, *PSMB8* expression and *PSMB9* expression highly correlated (Spearman R = 0.81, $P < 2.5E-6$). Of the 26 samples for which gene expression/mutation data and response were available, we considered

21 samples with a tumor purity >45% (12 responders and 9 non-responders). Importantly, our results did not depend on purity thresholds (Supplementary Table 13). Notably, mutational load, IFNγ signature, and T-cell infiltration were not associated with an improved response in this cohort (Fig. 5c). However, the tumors derived from patients who had a durable clinical benefit showed higher immunoproteasome subunit expression than those with no durable benefit ($P = 0.03$, Wilcoxon rank sum test), while it is less so for mutational load ($P = 0.1$, Wilcoxon rank sum test). Individual immunoproteasome subunit expression showed less significant association than did expression of both PSMB8 and PSMB9 ($P = 0.03$ for PSMB8 and $P = 0.06$ for PSMB9, Wilcoxon rank sum test), but the expression of the constitutive proteasome subunits showed no correlation ($P = 0.5$, Wilcoxon rank sum test) (Fig. 5c). We confirmed that immunoproteasome subunit expression strongly correlated with ICT response when tumor purity and the expression of constitutive proteasome subunits, mutational load, IFNγ, and CD8+ T-cell abundance were controlled for (Kendall tau = 0.33, $P < 0.06$), again pointing to the independent contribution of IP subunits to the ICT response. Furthermore, immunoproteasome subunit expression distinguished the responders from non-responders with a decent prediction accuracy (AUC of ROC = 0.75; Fig. 5d), the highest among mutational load, IFNγ signature, CD8+ T-cell abundance, and PDL1 expression, reaching AUC = 0.79 when combined with mutational load. This relationship remained robust for all 26 samples without the purity cut-off (Supplementary Table 13). The majority of high-purity (purity > 45%) tumors (6/8) exhibiting high immunoproteasome subunit expression and high mutational load (>50th percentile) were from patients who had a durable clinical benefit from the ICT, but notably, here, too, only a few of the patients (3/7) with low immunoproteasome subunit expression and low mutational load (<50th percentile) benefited from pembrolizumab or nivolumab.

## Discussion

This is the first report to establish the association between the overexpression of immunoproteasome subunits PSMB8 and PSMB9 and improved survival and enhanced response to immune-checkpoint inhibitors (both anti-CTLA4 and anti-PD1) in melanoma patients. By analyzing datasets from patient cohorts and performing systematic HLA peptidomic and immunoreactivity analyses of cells overexpressing the immunoproteasome subunits, we demonstrate that the overexpression of PSMB8 and PSMB9 results in enhanced reactivity of TILs toward melanoma cells, as a consequence of an altered repertoire of presented antigens. Indeed, the melanoma-TIL co-culture experiment, which evaluates the complete repertoire of presented peptides, takes into account the factors that affect the immune recognition —the level of presentation of the peptides, their binding, the abundance of T-cells and their activation state. Our analysis recapitulates previous studies showing that immunogenic peptides may be processed by both the constitutive proteasome and the immunoproteasome[12,14]. Notably, however, our data suggest that enhanced IP activity in tumors plays an important role in the improved immune response. The presence of the immunoproteasome-generated immunogenic antigens is likely to attract immune cells, as shown by our CIBERSORT analysis. These changes are accompanied by high cytolytic activity (as demonstrated by CYT score analysis) and lysis of melanoma cells by the secretion of granzyme and perforin[4]. This superior immune response in the tumor may underlie the observed higher survival rates of melanoma patients with high levels of the immunoproteasome subunits and an improved response to checkpoint inhibitors. Notably, the checkpoint datasets that are

publicly available are still of modest size, calling for further studies in extended cohorts. Remarkably, however, the combination of IP expression with tumor mutational load predicts anti-CTLA4 response with almost 0.9 AUC and anti-PD1 response with an AUC of almost 0.8. The insights gathered through these analyses suggest that the expression levels of immunoproteasome subunits PSMB8 and PSMB9 can serve as biomarkers for predicting survival of melanoma patients and for identifying patients likely to favorably respond to immune-checkpoint inhibitors.

## Methods

**TCGA melanoma tumor data**. We downloaded TCGA[32] gene expression and clinical profiles of 472 melanoma patients from the Genomic Data Common (GDC) TCGA data portal (https://portal.gdc.cancer.gov). (Supplementary Table 1).

**Patient survival analysis**. We performed two different analyses to identify the association of immunoproteasome genes with patient survival: Kaplan–Meier analysis and Cox proportional hazard model. We compared the survival of patients with high PSMB8 and PSMB9 (top tertile; $N = 155$) vs low PSMB8 and PSMB9 (bottom tertile; $N = 156$) using logrank test[33], and the effect size were quantified by the difference in the median survival time. To control for potential confounders, we performed a Cox regression analysis, while controlling for patients' age, sex, race and tumor purity.

$$h_s(t, \text{patient}) \sim h_{0s}(t)\exp\left(\beta_{\text{IP}}\text{IP} + \beta_a\,\text{age} + \beta_p\,\text{purity}\right), \quad (1)$$

where $s$ is an indicator variable over all possible combinations of patients' stratifications based on race and sex. $h_s$ is the hazard function (defined as the risk of death of patients per time unit), and $h_{0s}(t)$ is the baseline-hazard function at time $t$ of the $s$th stratification. The model contains three covariates: (i) IP: IP subunits expression, (ii) age: age of the patient, and (iii) tumor purity: the cancer cell fraction in the bulk tumor samples[18,19]. The $\beta$ s are the regression coefficients of the covariates, which quantify the effect of covariates on patients' survival, determined by standard likelihood maximization of the model[34]. The association between patient survival and other variables including CD8A, CD4, CD3G, IFNG, PSMB5, and PSMB6 expression levels, IFNγ signature (summed expression levels of IFNG, JAK1, JAK2, STAT1, STAT2), CD8+ T-cell abundance[20], and mutational load were evaluated by replacing the IP variable in Formula (1) with each of these variables. The abundance of 22 immune cell types was estimated using CIBERSORT[20]. For these variables, Cox regression analysis was performed with and without purity as a covariate. A multivariate Cox regression analysis was performed that incorporates IP subunit expression, CD8A, CD4, and CD3G expression, IFNγ signature, CD8+T-cell abundance, and mutational load altogether as independent variables.

**Cytolytic activity**. Cytolytic activity (CYT score) was calculated as described before ref. [4] for all melanoma samples in the TCGA dataset ($N = 472$). The expression levels of PSMB8 and PSMB9 were divided to low and high expression according to the average expression of each gene.

**PSMB8 and PSMB9 expression in tumor vs healthy tissue samples**. We downloaded transcriptomics data of 469 TCGA melanoma patients and 517 GTEx skin samples via UCSC Xena browser (http://xena.ucsc.edu), where the transcriptomics data was normalized with the exact same pipeline to facilitate the comparison between cancer vs healthy tissues. We compared the expression of PSMB8 and PSMB9 in these cancer vs healthy tissue samples using Student's $t$ test.

**Cells**. Cell lines 12T and 108T and their TILs were derived from pathology-confirmed metastatic melanoma tumor resections collected from patients enrolled in institutional review board (IRB)-approved clinical trials at the Surgery Branch of the National Cancer Institute. A375 is a commercial cell line that was purchase from ATCC. Whole exome sequencing (WES) of 108T and 12T was performed as described previously[35] and available in dbSNP under accession 1062266. The mutation lists are in Supplementary Tables 14, 15. Patient information including the HLA haplotype of these patients is found in Supplementary Table 16. EBV-transformed B-cells were purchased from the IHWG Cell and DNA Bank. Hybridoma cells HB95 were purchased from the ATCC and were used to purify pan-HLA-I antibodies for the preparation of the HLA affinity columns. All cell lines were tested regularly and were found negative for mycoplasma contamination (EZ-PCR Mycoplasma Kit, Biological Industries). Cells were authenticated by Finger printing with STR profiling (Panel: PowerPlex_16_5Nov142UAGC, Size: GS500 x35 ×50 ×250, Analysis Type: Fragment (Animal), Software Package: SoftGenetics GeneMarker 1.85).

**Stable expression of immunoproteasome subunits in melanoma cell lines**. Human PSMB8 cDNA was cloned into the pCDF1-MCS2-EF1-Puromycin vector

and human *PSMB9* cDNA was cloned into the pCDH1-CMV-MCS-EF1-Neomycin vector (Systems Biosciences). To produce lentivirus, the constructs were cotransfected into HEK293T cells with the pVSV-G and pFIV-34N helper plasmids. Virus-containing medium was collected 72 h after transfection, and filtered, aliquotted and stored at −80 °C. 108T, 12T and A375 cells were grown in RPMI-1640 supplemented with 10% FBS. Lentivirus for *PSMB8* and empty vector control were used to infect the cells, and after stable expression of the *PSMB8* protein was determined by western blot analysis, the cells were then infected with the lentivirus for *PSMB9* and empty vector respectively to produce cells with double transfection of *PSMB8* and *PSMB9* or two empty vectors. Each cell line was infected three times in order to receive three independent cell cultures of both double expression of the immunoproteasome subunits or empty vector control. Cell pellets of cells with overexpression or empty vector control were collected from $2 \times 10^8$ cells, and in total three different experimental replicates for each condition and cell line were collected.

**Immunoblotting**. To validate the stable infection of *PSMB8* and *PSMB9*, cells were gently washed two time in PBS and then lysed in sample buffer 2×. The extracts were sonicated (50 W, $2 \times 7$ s), incubated on ice for 15 min, and boiled in 95 °C for 5 min. The samples were then subjected to 10% SDS-PAGE. Immunoblots were probed with anti-PSMB8 (13726, Cell Signalling), anti-PSMB9 (3328, Abcam) and anti-GAPDH (MAB374, Millipore). Blots were developed with HRP-conjugated anti-mouse or ant-rabbit Abs, using SuperSignal West Pico Chemiluminescent Substrate or SuperSignal West Femto Chemiluminescent Substrat from Thermo Scientific (Waltham, MA, USA). Pictures of the blots were taken using BioRad ChemiDoc MP system and figures prepared by using Image lab (BioRad). PSMB8 and PSMB9 were blotted from same cell lysate on different blots due to the proximity in their size.

**IFNγ treatment**. 12T and 108T cells were treated with 250 U/ml IFNγ (Peprotech) for 48 h to induce a maximal expression of HLA molecules with minimal cell mortality. Three different experimental replicates of treated and non-treated cells were collected to cell pellets of $2 \times 10^8$ cells.

**Flow cytometry**. $5 \times 10^6$ 12T and 108T cells with immunoproteasome overexpression and empty vector control, and treated or non-treated with IFNγ were collected and washed with PBS. Then cells were incubated with PE/Cy7 anti-HLA-A, B, C (W6/32) antibody (311429, Biolegend) for 30 min on ice. Cells were later washed twice with PBS and analysed using BD LSR II flow cytometer (BD Biosciences). Data were analysed using the FlowJo software.

**Proteasome activity assay**. $10^7$ cells with overexpression or empty control cells were collected and washed twice with PBS. Cell pellets were later re-suspended in lysis buffer (50 mM TRIS pH 8 and 0.5% NP-40). Lysates were passed 10 times through a 28 G needle, incubated on ice for 15 min, and then centrifuged at 16,000g for 30 min. Protein concentration was measured using Pierce$^{TM}$ BCA protein assay kit (Thermo Scientific). 20 μg of cellular lysate was incubated in reaction buffer (50 mM HEPES pH8, 5 mM MgCl$_2$, 2 mM ATP, 1 mM DTT) and 0.1 mM suc-LLVY-AMC or ac-PAL-AMC (S-280 and S-310, Biotest). Fluorescence levels were measured every minute for 3.5 h, using the Typhoon-9410 laser flatbed scanner (GE Healthcare, USA) (Excitation: 360 nm, Emission: 460 nm). Background protease activity was determined for each condition from an identically prepared sample with the addition of 0.04 mM MG132 proteasome inhibitor (474791, Calbiochem). Each measurement was performed in four replicates. RFUs were plotted over time and another graph showing the RFU between starting and end time points.

**Purification of membrane HLA molecules**. For the HLA peptidomics analysis we used three experimental replicates per each cell line and each condition (empty vector/ overexpression and treated/ non-treated cells). Samples were processed as described previously[23,25,36]. Briefly, cell pellets were lysed with lysis buffer containing 0.25% sodium deoxycholate, 0.2 mM iodoacetamide, 1 mM EDTA, 1:200 protease inhibitors cocktail (Sigma-Aldrich), 1 mM PMSF and 1%octyl-b-D glucopyranoside in PBS, and then incubated at 4 °C for 1 h. The lysates were cleared by centrifugation at 4 °C and 48,000g for 60 min, and then passed through a pre-clearing column containing Protein-A Sepharose beads.

HLA-I molecules were immunoaffinity purified from cleared lysate with the pan-HLA-I antibody (W6/32 antibody purified from HB95 hybridoma cells) covalently bound to Protein-A Sepharose beads (Thermo Fisher Scientific, as in ref. [23]). Affinity column was washed first with 10 column volumes of 400 mM NaCl, 20 mM Tris–HCl, pH 8.0 and then with 10 volumes of 20 mM Tris–HCl, pH 8.0. The HLA peptides and HLA molecules were eluted with 1% TFA followed by separation of the peptides from the proteins by binding the eluted fraction to disposable reversed-phase C18 columns (Harvard Apparatus) as in ref. [37]. Elution of the peptides was done with 30% acetonitrile (ACN) in 0.1% trifluoracetic acid (TFA). The eluted peptides were cleaned also by C18 stage tip[38].

**Identification of the eluted HLA peptides**. The HLA peptides were dried by vacuum centrifugation, re-solubilized with 0.1% formic acid and resolved on

capillary reversed-phase chromatography on 075 × 300 mm laser-pulled capillaries, self-packed with C18 reversed-phase 3.5 μm beads (Reprosil-C18-Aqua, Dr. Maisch GmbH, Ammerbuch-Entringen, Germany[39]). Chromatography was performed with the UltiMate 3000 RSLCnano-capillary UHPLC system (Thermo Fisher Scientific), which was coupled by electrospray to tandem mass spectrometry on Q-Exactive-Plus (Thermo Fisher Scientific). The HLA peptides were eluted with a linear gradient over 2 h from 5 to 28% acetonitrile with 0.1% formic acid at a flow rate of 0.15 μl/min. Data was acquired using a data-dependent "top 10" method, fragmenting the peptides by higher-energy collisional dissociation (HCD). Full scan MS spectra was acquired at a resolution of 70,000 at 200 m/z with a target value of $3 \times 10^6$ ions. Ions were accumulated to an AGC target value of $10^5$ with a maximum injection time of generally 100 msec. The peptide match option was set to Preferred. Normalized collision energy was set to 25% and MS/MS resolution was 17,500 at 200 m/z. Fragmented m/z values were dynamically excluded from further selection for 20 s. The MS data were analyzed by MaxQuant[40] version 1.5.3.8. Peptide was searched against the UniProt human database, and customized reference databases which contained the mutated sequences identified in the sample by WES. N-terminal acetylation (42.010565 Da) and methionine oxidation (15.994915 Da) were set as variable modifications. Enzyme specificity was set as unspecific and peptides FDR was set to 0.05. The match between runs option was enabled to allow matching of identifications across the samples belonging the same patient. HLA peptidomics data have been deposited to the ProteomeXchange Consortium[41] via the PRIDE[42] partner repository with the dataset identifier PXD015957.

**Analysis of differentially presented peptides**. Peptides identified through MaxQuant were first filtered to remove reverse sequences and known contaminants. Gibbs clustering was used to see if the peptides cluster according to the cells' HLA haplotype and the cluster name was assigned for each peptide in the table. Similarly, using NetMHCpan 4.0[43,44], peptides were analyzed to see if they are predicted to bind the cells' HLA haplotype, the binding prediction score was assigned to the peptide only if the rank <2%. Peptides were further used for the differential presentation analysis if they were assigned with HLA allele by both the Gibbs clustering and NetMHCpan predictions. Neo-antigens and TAAs were marked in a designated column. For the differential presentation analysis, graphics and statistical analysis we used the Perseus computational platform[45] version 1.6.6.0. Peptide intensities were Log-2 transformed and missing intensity values were imputed by drawing random numbers from a Gaussian distribution with a standard deviation of 20% in comparison to the standard deviation of the measured peptide abundances. We excluded sample A375 OE3 from the analysis as lower number of peptides were identified in it compare to the other samples in the triplicate, as well as 108T OE2 as it was different from the two other replicates. Volcano plots which show differentially presented peptides of the relative intensities of HLA peptides between cells with overexpression of immunoproteasome subunits and empty vectors were created, as well as cells that were treated and non-treated with IFNγ. The x axis represent the Log2 fold changes of the peptide intensities, and the y axis represent the significance levels calculated by two-sided unpaired t test with a FDR of 0.05 and S0 of 1. The peptides that were found to be significantly differentially presented between the overexpressing cells and the empty vector control (or the treated vs. non-treated cells) were used for further analysis of peptides' immunogenicity assay.

**Gibbs clustering**. In order to classify the clustered peptides into HLA alleles, we first identified for each allele its motif. For that we retrieved all HLA-I epitopes registered under this allele from the Immune epitope database[46] (IEDB, www.iedb. org, as of July 2019). All peptides were annotated as positive in "MHC ligand assays" to the specific HLA-I allele and between 8 and 13 amino acids long. The GibbsCluster 2.0 server[47] (www.cbs.dtu.dk/services/GibbsCluster) was used to align the peptides using the "MHC class I ligands of length 8–13" parameters, number of clusters was set to one and the trash cluster option was disabled.

Each set of peptides (patient and treatment/ overexpression) was also clustered using Gibbs clustering, with the "MHC class I ligands of length 8–13" parameters, number of clusters was set to six and the trash cluster option was enabled. All motifs were generated by Seq2Logo 2.0[48] (http://www.cbs.dtu.dk/biotools/Seq2Logo) with the default settings. Since the number of peptides per each allele is different, for alleles with a higher number of peptides (as the HLA-A alleles) the unbiased clustering resulted in more than one cluster for these alleles. In these cases, we added all clusters that matched the allele motif in the same category. The change in number of peptides to allele also resulted in clusters with mixed motifs that were similar. In these cases, we assigned the cluster to the allele that had the highest representation in the cluster and added a note to which other alleles are mixed within. We notice that most peptides that are not assigned to a specific cluster were also not predicted to bind any HLA allele by NetMHCpan and were longer. Those are probably unspecific contaminates that we excluded from all further analyses.

**Identification of TAAs**. We selected known cancer and melanoma antigens that were described previously in the Cancer/Testis database[26] (CTDatabase, http://

www.cta.lncc.br/), peptide database[27] (https://www.cancerresearch.org/scientists/events-and-resources/peptide-database) and a human melanoma dataset[28].

**Killing assay.** $5 \times 10^5$ 12T and 108T cells with overexpressing the IP subunits or empty vector control, and treated or non-treated cells with IFNγ were seeded a day before in 6 well plates. TILs were added to wells in effector to target ratios of 0:1, 0.25:1, 0.5:1 and 1:1 and incubated for 8 h or 12 h for 12T and 108T, respectively. Then plates were washed twice by PBS to remove TILs. Cells were collected and live cells were counted with trypan blue using the Countess II Automated Cell Counter (Thermo Fisher Scientific). Each measurement was performed in three biological replicates.

**Analysis of T-cell reactivity by IFNγ release assay.** To evaluate the immunogenicity of the identified peptides the identified peptides were synthesized, loaded on antigen presenting cells and co-cultured with the autologous TILs. All synthetic peptides were purchased from GeneScript as crude peptides. EBV-transformed B cells that express the correct HLA allele were loaded with the candidate peptides at a concentration of 10 μM for 2 h at 37 °C. Following three washing steps, the loaded B-cells were co-cultured with the autologous TILs in 1:1 ratio ($10^5$ cells) for an overnight incubation. The amounts of soluble IFNγ secreted from the TILs were measured by ELISA assay (ELISA MAX™ Deluxe Set Human IFNγ, Biolegend). Plates were scanned using the Typhoon-9410 laser flatbed scanner (GE Healthcare, USA) and analysed using MyAssays analysis software tool (www.myassays.com). Concentrations were calculated using four parameter logistic fit. From each peptide measurement we reduced the background measurement of the control, which was the same B-cells to which we added only DMSO and were later co-cultured with the TILs. All measurements were done in triplicates. A control peptide was used to normalize the concentration values between different ELISA plates of the same experiment. Graphs and statistics were done using GraphPad Prism 5.

**Predicting the effectiveness of immune-checkpoint inhibitors therapy.** We analyzed two different melanoma cohorts[2,31] treated with anti-CTLA4 therapy and anti-PD1 therapy. We compared the IP subunit expression between responders and non-responders using Wilcoxon ranksum test. The predictive power of the IP subunits, mutational load, IFNγ signature, cytolytic score[4], CD8+ T-cell abundance[20], and PDL1 expression (for anti-PD1 cohort only) for the success of immunotherapy was evaluated using ROC and precision-recall analysis with varying tumor purity[19] thresholds. Partial correlation analysis was performed using R library 'ppcor' to determine Kendall rank correlation between IP subunit expression and response to ICT while controlling for mutational load, tumor purity, IFNγ signature, and CD8+ T-cell abundance.

**Reporting summary.** Further information on research design is available in the Nature Research Reporting Summary linked to this article.

## Data availability
Data supporting the findings of this study are available within the paper and its Supplementary Information files. The source data underlying Supplementary Fig. 6 are provided as a Source Data file. HLA peptidomics data have been deposited to the ProteomeXchange Consortium via the PRIDE partner repository with the dataset identifier PXD015957.

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

## Acknowledgements

The authors would like to acknowledge the Smoler Proteomics center and Ilana Navon for performing the LC-MS/MS. We would also would like to thank Prof. Moshe Oren for constructive discussions. We thank Dr. Yifat Merbl for her help with reagents. This work was supported by the Intramural Research Programs of the National Cancer Institute. This project was sponsored by the ERC (CoG-770854), European Union's Horizon 2020 research and innovation programme (#754282), H2020 European Research Council (#712977), Israel Science Foundation (#696/17), MRA (#622106), Rising Tide Foundation (to Y.S.), Wagner-Braunsberg Family Melanoma Research Fund, Jean-Jacques Brunschwig Fund for the Molecular Genetics of Cancer, Comisaroff Family Trust, Meyer Henri Cancer Endowment, Ted and Sylvia Quint, Laboratory in the name of M.E.H. Fund established by Margot and Ernst Hamburger, Knell Family and Hamburger Family (to Y.S.).

## Author contributions

S.K., E.B., and M.A. preformed the HLA peptidomics experiments, S.K. analyzed the experimental data and preformed all the other assays. P.G. and G.Y. helped in growing cells for HLA peptidomics. J.S.L. and S.P. preformed the analysis of patient cohorts. R.L. and A.P. helped in statistical analyses. G.B.E., R.O. and L.B. provided technical help. E.B., Y.L., and A.A. advised in HLA peptidomics and preformed mass spectrometry analyses. S.A.R. and M.L. provided patient material. S.K., J.S.L., E.R., and Y.S. designed and directed the study and wrote the manuscript. Every author reviewed the work.

## Competing interests

The authors declare no competing interests.
