## [Peer Review File · Nature Communications]

Reviewers' comments:

Reviewer #1: HLA peptidomics
(Remarks to the Author):

Kalaora et al. present a study on the association of immunoproteasome (IP) expression as a proposed prognostic and predictive marker for melanoma patient survival and response to immune checkpoint inhibitor therapy. They aim to elucidate the underlying mechanism by analyzing changes in the HLA class I peptidome of 3 melanoma cell lines upon overexpression of IP subunits and stimulation with IFN- γ . Immunogenicity of tumor-associated selfantigens and 4 neo-antigens is assessed by IFN γ -ELISA of autologous TIL lines and the data is presented as evidence for increased immunogenicity of tumor antigens upmodulated upon IP overexpression. While the subject matter of identifying prognostic and predictive biomarkers for immunotherapy of melanoma and identifying the underlying mechanisms is a timely and highly relevant topic, the data and statistical analysis presented by Kalaora et al. do not support most of the claims. The data and the manuscript in its current form might only be considered for publication in Nature Communications after substantial major revision.

Major points

Novelty: It is well known that cancers with T-cell infiltration show a better response towards CKIs. T cells secrete IFN, which leads to upregulation of several genes, including LMP2 and LMP7. Accordingly, it is not surprising to see an association of immunoproteasome expression with melanoma patient survival as well. It would be interesting to learn if T cell infiltration is a result of increased immunoproteasome subunit expression or the reason for this upregulation. Most of the IFN-pathway genes should show the same association as described here, confounders such as T cell infiltration (e.g. CD8, CD4, CD3 expression) were not discussed in the statistical analysis shown in the body of the publication but in the supplementary part only (Cibersort, Suppl. Table 1).

The influence of IFN γ on peptide presentation by differences in proteasomal subunit usage has been shown before as cited by the authors themselves.

The chicken or the egg causality dilemma is not solved in this manuscript: T cell infiltration leading to IFN γ -release, IFN γ -induced immunoproteasome expression, IP-associated epitope presentation leading to IFN γ -release by T cells.

Focus on HLA peptidome part

Peptide sets used in the manuscript like "differentially presented peptides" or "immunogenic peptides", that are central for the provided argumentation, should be properly defined. In the current form, elucidation of those definitions was extremely cumbersome and only possible by working through the supplementary information.

The differential HLA peptidome analysis apparently was performed without any technical or biological replication. The reproducibility of the assay and statistical significance of differential MS1 signal intensities thus cannot be analyzed. In the current manuscript the authors simply consider any peptide with MS signal FC-values deviating from 1 as differentially presented, which is inappropriate. "Peptides differentially presented by IP OE cells" were apparently defined as peptide with either missing identification in EV or with OE/EV fold-change above 1 while "peptides differentially presented by EV cells" are either missing identification in OE or have a fold change below 1. This central limitation also renders the entire downstream analysis on differential immunogenicity of modulated peptides non-informative.

As conventionally employed in label-free quantification MS, the authors should conduct at least three experimental replicates per cell line (independent cell culture, transfection and HLA peptide isolation). LC-MS data acquisition should comprise technical replicates and be block randomized to control for potential spray instabilities and other technical variability which might not be corrected by normalization. Volcano Plots may then be shown to visualize the effect size and the significance levels of modulation. Quality metrics of the HLA isolations should be presented by stating peptide yields and purities, e.g. as percentage of predicted binders or as percentage of clustered peptide

vs. outliers in Gibbs clustering. Gibbs clustering parameter need to allow as many clusters as there are HLA allotypes expressed by the sample. The trash cluster option should be enabled. Presenting only a single Gibbs cluster for a class I peptidome comprising 6 HLA allotypes is not sufficiently informative. Peptides without HLA assignment or source cluster may be excluded from the differential HLA peptidome analysis as they likely constitute unspecific contaminants (for instance LDTNADKQLS which is present in every cell-line). The HLA typing of the cell lines should be provided, the MS raw data should be made publicly available.

Because of their high potential impact (Bassani-Sternberg et al. Nat. Commun. 2016), putative IDs from neo-antigen need to be validated by calculating the spectral correlation with their synthetic counterparts and by co-elution with stable isotopically labeled peptides (see Blatnik et al., Proteomics 2018 [10.1002/pmic.201700390]).

The hydrophobicity scoring method is not described, the effect size appears to be minute. Differences in peptidome hydrophobicity profiles (Fig. 2) may also arise from different degrees of adsorption to autosampler vials due to different sample standing time prior to LC-MS as described by the CPTAC consortium (Hoofnagle et al., Clin Chem. 2016). The method section therefore should accurately describe sample handling, replication and statistical analysis as outlined in the Minimal Information about an Immuno-Peptidomics Experiment (MIAIPE) guidelines (Lill et al., Proteomics 2018) and as required by the reporting summary by Nature Communications.

No reproduction is shown for the IFN γ -ELISAs. For all 3 cell lines prevalence and intensity of IFN γ release is highly similar, irrespective of differential peptide presentation. The conclusion drawn from these data is heavily influenced by few outliers. These experiments require reproduction and adequate statistical evaluation to substantiate the association of differential immunogenicity of peptides associated with IP-overexpression. The method section should accurately describe whether ELISAs were performed in parallel on the same plate for the different peptide populations. In addition, based on the provided summaries it can be deduced that "immunogenic peptides" were defined as peptides with IFN γ > 0 pg/ml which does not reflect the background signals observable in such assays.

It should be discussed if ligands restricted to HLA-A*02 are expected to be more immunogenic than those presented on e.g. HLA-A*03, because they are more hydrophobic due to their anchor amino acid usage.

Minor points:

- The claim of novelty ("We show here for the first time that the overexpression of these subunits is correlated with improved survival and better response to immune checkpoint inhibitors.") should be toned down. Tripathi et al have previously shown similar results in NSCLC [10.1073/pnas.1521812113]. In general, the limitation of the findings to the utilized melanoma model should clearly be stated.

- Figure 1A: the legend for high and low IP expression is swapped

- The statistical methods on the identification of PSMB8 and PSMB9 as a prognostic marker should be better explained:

- o If for z-transformation only the melanoma tumors were used, the finding that 15% are overexpressed at z-score > 1 is an implicit consequence of the standard-normal-distribution of the Z-score (see 68–95–99.7-rule) and does not confirm over-expression.

- o If for z-transformation the mean and standard deviation was computed based on another reference data set (skin normals or other healthy tissues), this data set should be described and proper over-expression statistics (e.g. t-test) should be provided.

- o

- The Cox proportional hazards model was only applied to the dichotomized variable based on an arbitrary cutoff (top vs bottom 10%) which results in potential bias (See Eng et al. [10.18632/oncotarget.6121]). Thus, additional cutoffs should be displayed in the Kaplan-Meier plot and the Cox Hazard ratio and p-Value should be shown for the original continuous variable. It should also be shown for the mutational load.

- o The deltaAUC is not commonly used to quantify effect size in clinical research. Median OS should be shown instead.
- o Table S1: Instead of raw coefficients (beta), hazard ratios (HR) should be provided based on continuous expression values after z-transformation. Thus, HRs will be comparable between variables and be associated with change of 1 standard deviation rather than change of 1 unit of expression.
- Hydrophobicity data: As stated above, experimental confounders need to be precluded and the utilized score should be described. The correlation of the score with the experimental RTs should be analyzed. The statistics on the comparison of avg. hydrophobicity should be revised: The hydrophobicity scores are not normally distributed. The relevance of the rather small differences is questionable.
- The numbers of Exome-seq based mutations for the 3 melanoma cell lines should be provided. Identified NeoAg should be displayed separately in Figure 3A to clearly display their relative contribution.
- Figure 4: Different scales are used for the intensity ratio. The coloring not symmetrical for up- and down modulation.
- Computing the arithmetic mean of PSMB8 and PSMB9 is questionable if just Spearman correlation as a non-parametric statistic is shown. A scatterplot of both measurements and linear modelling should be provided to justify this step.
- Not all authors are listed in the contribution section.

Reviewer #2: Immunoproteasome and biomarkers
(Remarks to the Author):

The manuscript by Kalaora et al addresses one of the largest outstanding questions in oncology – factors behind the heterogeneous response to immune checkpoint inhibition. This work focuses on a relatively new target, the immunoproteasome, as a predictor of this response. This hypothesis makes sense, in that antigen presentation, especially of tumor related neoantigens, is central to interaction between tumor cells and cytotoxic T-lymphocytes. The paper uses a variety of approaches, including analysis of TCGA data, cell culture models, and proteomic analyses of MHC-eluted peptides, to support their central hypothesis.

The paper is well written and I believe should be accepted with minor revision. The conclusions are novel, and builds on previous work showing altered immune responses with immunoproteasome expression in melanoma as well as differential expression of the immunoproteasome in cancer. This is supported, mostly, by the data presented. It will have a good impact on the field and potentially alter treatment and outcomes.

Issues

- The TCGA analyses would be stronger if, from the outset, the authors made clear the tumor purity. Additionally, while the authors quantitate TILs and inflammatory cells, the immunoproteasome is expressed at high levels in antigen presenting cells including dendritic cells. While I see that other infiltrating and stromal cells are quantitated, is it possible to estimate APCs in these tumor specimens? If the immunoproteasome is predominantly in APCs, this could drastically alter the conclusions drawn in the paper. These tumors likely have higher levels of IFN-gamma, and thus could have a highly altered tumor microenvironment as well.
- In the TCGA data, is there any correlation with PSMB10 expression? PSMB10, as well as some members of the TAP, have roles in immunoproteasome assembly and function.

- I would like to see confirmation of expression of the immunoproteasome subunits in the overexpression cell lines. It would be useful to also have confirmation of enzymatic activity, perhaps through a commercially available luminescence assay. This would help understand what percentage of proteasome function in these overexpression cells is due to constitutive proteasome and which is due to immunoproteasome.
- Is there any quantitation of MHC expression, both in TCGA tumor specimens and in the IFN-gamma treated cells? Other components of the antigen presenting machinery are known to be highly induced by IFN-gamma.
- A sentence or two describing the nature of the TAAs and neo-antigens examined would be useful. Were these point mutations? Were they deletions or frameshifts? Was there differences in presentation based on the mutation type?
- In figures 4 and 5, I would appreciate a better discussion of the statistical methods used to judge statistical significance of IFN-gamma treatment. By dotplot, these looks to be driven by one or two outliers in each of these experiments. For the vast majority, IFN-gamma seems to be at the limit of detection.

Minor issues

- A few typos. In the "Cells" section of the methods line 152 "obtain" -> "obtained"
- Figure 4/5: The blue/black color scheme makes it difficult to discern which antigens are more immunogenic.
- Figure 6: the green boxplot in figure A is truncated

Edwin Ostrin

Reviewer #3: Melanoma Immunotherapy (Remarks to the Author):

In this manuscript, Kalaora and colleagues advance an attractive hypothesis that the immunoproteasome subunits, which are IFN/stress inducible and alter the antigen presentation machinery, are differentially expressed in tumor cells and may contribute to shaping the anti-tumor immune response via immunoproteasome-specific neo-antigens.

They present the following pieces of data to support their hypothesis:

1. TCGA analysis of melanoma patients analyses by RNAseq for immunoproteasome expression - the top 10% vs bottom 10% show a difference in survival (which also corresponds to difference in "CYT" score or T cell infiltrate pattern).
- 2 MS and in vitro T cell stimulation assays on 3 melanoma cell lines (with matched blood) to show enhanced "immunogenicity" of immunoproteasome expressing tumors (which generate unique antigens).
3. Analysis of datasets of pts with melanoma treated with either CTLA-4 or PD-1 blockade showing that immunoproteasome expression (but not proteasome expression) is associated with better outcomes.

The main issue that the authors have not addressed is how important the IP expression is for influencing clinical outcomes - i.e. is increased elevation in IP levels simply a reflection of a more inflamed tumor environment that has already been correlated with better outcomes (IP is IFN induced). In that vein, it would be helpful to see a broader analysis of IFN-g responsive genes and a comparison to existing gene-signatures that have already been linked to clinical outcome in similar populations. In this dataset, are there "exceptions" that prove the rule - i.e. a tumor where IFN-g induced genes are upregulated, but IP subunits are lost or are not upregulated and the

patient is resistant to checkpoint blockade ?

Even with this complete analysis, challenging to causally link IP expression to clinical outcome using retrospective analysis of banked samples, and the authors should take care in describing the clinical data as exploratory and hypothesis generating.

Reviewer response

We thank the reviewers for their comments regarding our manuscript. We have addressed their concerns following the guidelines outlined in our correspondence and discussion, as described in detail below. The revised manuscript includes a number of key new genetic, biochemical and cellular experiments that further support and strengthen our results.

In particular, the revised manuscript now includes the following new experiments:

1. A more comprehensive analysis of the patient data that takes into account additional possible confounders that might affect the RNA expression level of the immunoproteasome subunits in the tumor samples. We included in this analysis the effect of the IFN γ signature and a computational estimation of CD8 $^+$ T-cell infiltration. We further preformed a similar analysis to exclude APCs as a confounding factor for the immunoproteasome expression level.
2. We show significant associations of immunoproteasome expression with patient survival, which remain in high purity samples. This is in contrast to the associations of IFN γ signature and computationally estimated CD8 $^+$ T-cell infiltration with patient survival, which are lost in high purity samples.
3. We show that the level of expression of the immunoproteasome subunits has a better predictive power compared to known correlates of checkpoint inhibitors commonly used today, such as mutational load, IFN γ levels and T-cell infiltration.
4. We performed new killing experiments using patient melanoma cells overexpressing the immunoproteasome subunits and the patient's autologous tumor-infiltrating lymphocytes. These experiments further testify that the T-cells have an enhanced response against cells overexpressing the immunoproteasome subunits.
5. We repeated all HLA peptidomics analyses in triplicates and showed the reproducibility of our results. We filtered peptides that do not match the patient's HLA alleles using both NetMHCpan and Gibbs clustering. We also performed a more strict selection process of the differentially presented peptides after immunoproteasome overexpression and better presented the results through the use of volcano plots.
6. We preformed several validation experiments, including the following: (1) Proteasome activity assay, to validate that overexpressed immunoproteasome subunits were incorporated into the proteasome complex and are active. (2) Flow cytometry analysis, to determine the HLA expression in cells after IFN γ treatment and overexpression of the immunoproteasome subunits. (3) Western blot analysis, to check the expression levels of immunoproteasome subunits in cells after IFN γ treatment.

7. We validated the identification of neo-antigens by comparing their spectra with their synthetic counterparts and by spiking the sample with a stable isotopically labeled peptide to show that they co-elute.
8. We added new tables containing patient data, including their HLA typing and mutation lists.
9. We deposited the MS data in a public repository.

Taken together, we feel our additional results further establish and clarify the association between the expression of the immunoproteasome subunits with better prognosis and response to checkpoint therapies in melanoma patients.

Reviewers' comments:

Reviewer #1: HLA peptidomics (Remarks to the Author):

Kalaora et al. present a study on the association of immunoproteasome (IP) expression as a proposed prognostic and predictive marker for melanoma patient survival and response to immune checkpoint inhibitor therapy. They aim to elucidate the underlying mechanism by analyzing changes in the HLA class I peptidome of 3 melanoma cell lines upon overexpression of IP subunits and stimulation with IFN- γ . Immunogenicity of tumor-associated selfantigens and 4 neo-antigens is assessed by IFN γ -ELISA of autologous TIL lines and the data is presented as evidence for increased immunogenicity of tumor antigens upmodulated upon IP overexpression. While the subject matter of identifying prognostic and predictive biomarkers for immunotherapy of melanoma and identifying the underlying mechanisms is a timely and highly relevant topic, the data and statistical analysis presented by Kalaora et al. do not support most of the claims. The data and the manuscript in its current form might only be considered for publication in Nature Communications after substantial major revision.

Major points

1. Novelty: It is well known that cancers with T-cell infiltration show a better response towards CKIs. T cells secrete IFN, which leads to upregulation of several genes, including LMP2 and LMP7. Accordingly, it is not surprising to see an association of immunoproteasome expression with melanoma patient survival as well. It would be interesting to learn if T cell infiltration is a result of increased immunoproteasome subunit expression or the reason for this upregulation. Most of the IFN-pathway genes should show the same association as described here, confounders such as T cell infiltration (e.g. CD8, CD4, CD3 expression) were not discussed in the statistical analysis shown in the body of the publication but in the supplementary part only (Cibersort, Suppl. Table 1).

The influence of IFN γ on peptide presentation by differences in proteasomal subunit usage has been shown before as cited by the authors themselves.

The chicken or the egg causality dilemma is not solved in this manuscript: T cell infiltration leading to IFN γ -release, IFN γ -induced immunoproteasome expression, IP-associated epitope presentation leading to IFN γ -release by T cells.

Response: Thanks for the insightful comment. As the reviewer noted in the summary paragraph, we “present a study on the association of immunoproteasome (IP) subunit expression as a proposed prognostic and predictive marker for melanoma patient survival and response to immune checkpoint inhibitor therapy.” Following the reviewer’s comment, we performed a systematic analysis comparing the performance of *IP subunit expression* to that of *T-cell infiltration* or *IFN γ signature* in predicting patient survival and the response to immune checkpoint therapy.

To this end we first extended our analysis on TCGA melanoma cohort from 356 samples in the original manuscript to 472 samples in the revised manuscript, which recently became available through the updated Genomic Data Common (GDC) TCGA data portal (<https://portal.gdc.cancer.gov>). Our patient survival analysis in this extended data shows that the immunoproteasome expression is associated with patients’ prognosis independent of the levels of IFN γ or T-cell infiltration.

- (1) As the reviewer pointed, we indeed observed that IFN γ signature, IFNG expression, computationally estimated CD8+ T-cell infiltration¹, and T-cell related genes (e.g. CD8A) show significant associations with patient survival (Figure R1A,B,C,D). However, notably, this signal becomes insignificant when tumor purity is controlled for in the Cox model (Table R1).
- (2) In contrast, in the case of immunoproteasome, Both for PSMB8 and PSMB9, their expression is associated with better prognosis (Figure R1 E,F) and this association remains significant even when the tumor purity is controlled for (Table R1). Taken together with (1), these results indicate that IP expression significantly contributes to the survival independent of tumor purity, while IFN γ signature, IFNG expression, CD8A expression or CD8+ T-cell infiltration does not.
- (3) Finally, when we construct a multivariate Cox model combining IP expression (summed expression of PSMB8/9) together with IFN γ signature and T-cell infiltration, we see a significant association of IP expression with patient survival, but we do not see such associations for IFN γ or T-cell infiltration. Taken together, these results indicate that IP subunits are significantly associated with patient survival and this association is likely to be independent of IFN γ or T-cell infiltration (Table R1).

A

B

C

D

E

F

Figure R1. Comparative Kaplan Meier (KM) analysis of IFN γ , T-cell infiltration, and IP expression in TCGA melanoma patients. The KM plots compare the survival of patients with high (top tertile, blue) vs low (bottom tertile, yellow) levels of (A) IFN γ signature, (B) IFNG expression, (C) estimated CD8+ T-cell abundance, (D) CD8A expression, (E) PSMB8 expression, and (F) PSMB9 expression. For each variable, the survival of patients with top tertile vs bottom tertile was compared using logrank test, and the p-values are denoted in the figure. The table on the bottom summarizes the patients at risk in different time points, and the median survival differences are 4025, 4025, 3115, 3738, -783, 3738, and 4025, respectively.

	Univariate model		Univariate model + purity		Multivariate model	
	Hazard ratio	Wald P	Hazard ratio	Wald P	Hazard ratio	Wald P
IFN γ signature	0.709	0.000964	0.763	0.094754	1.042	0.861773
IFNG expression	0.691	0.001131	0.745	0.087037	1.116	0.679433
CD8+T-cell	0.766	0.015118	0.877	0.319634	1.097	0.576941
CD8A	0.694	0.000629	0.706	0.063550	0.886	0.628285
CD4	0.682	0.000536	0.678	0.078271	0.721	0.122652
CD3G	0.695	0.000953	0.695	0.114972	0.871	0.627104
Mutational load	0.796	0.074994	0.806	0.089139	0.842	0.194167
PSMB8	0.683	0.001128	0.753	0.036950	-	-
PSMB9	0.649	8.59E-05	0.651	0.009516	-	-
PSMB8/9	0.646	0.000138	0.677	0.011117	0.646	0.045741

Table R1. Comparative Cox regression analysis of IFN γ , T-cell infiltration, and IP expression in TCGA melanoma patients. The table shows the hazard ratio and corresponding Wald p-values of Cox univariate and multivariate models with IFN γ signature, IFNG expression, computationally estimated CD8+ T-cell infiltration, T-cell related gene expression (CD8A, CD4, and CD3G), PSMB8, and PSMB9 expression, and the summed expression of PSMB8/9 as independent variables for explaining patient survival.

To further establish this line of inquiry, we checked the correlation between IP expression and the expression of IFN γ -related (Figure R2A) or T-cell related genes (Figure R2B) in high vs low purity tumors. We observe that the correlation between IP subunits and IFN γ genes or T-cell related genes is significantly reduced in high purity tumors compared to that in low purity tumors (Figure R2). This suggests that the IP subunit expression *in the cancer cells* (as reflected in the high purity tumors) are not mere consequence of IFN γ release or T-cell infiltration, but rather independent of IFN γ release or T-cell infiltration.

Figure R2. The correlation between IP and IFN γ related gene expression is lost in high purity tumors. The Y-axis shows the correlation between the IP subunit expression and the expression of (A) IFN γ -related genes and (B) T-cell related genes in high (yellow; top 10-percentile) vs low (gray; bottom 10-percentile) tumors.

We next compared the prediction accuracy for response to immune checkpoint therapies of (i) immunoproteasome expression to that of (ii) IFN γ signature, and (iii) T-cell infiltration. Our analysis shows that the prediction accuracy of IP subunits is the highest in high purity tumors (Figure R3), while it is not the case in low purity tumors (Figure R4). In combination with mutational load, IP subunit expression provides the highest prediction accuracy in both datasets of anti-PD1 and anti-CTLA4. This supports the notion that the IP subunit expression *in cancer cells* is predictive of immune checkpoint therapies response.

Figure R3. IP subunits predicts the success of immune checkpoint therapy in high purity tumors. The Receiver Operating Curves (ROC) show that (A) IP expression itself shows the highest prediction accuracy for anti-PD1 therapy followed by mutational load, PDL1 expression, IFN γ signature, and CD8+ T-cell abundance. (B) IP expression shows the highest prediction accuracy for anti-CTLA4 therapy when combined with mutational load, followed by CD8+ T-cell abundance, IFN γ signature, and mutational load. We used the samples with tumor purity greater than 0.45, and this threshold was used throughout the paper.

Figure R4. IP subunits predicts the success of immune checkpoint therapy in high vs low purity tumors. The AUC of ROC curves (y-axis) is presented for (A) anti-PD1 and (B) anti-CTLA4 therapy for different variables IP subunit expression, mutational load, and their combination, PDL1 expression, IFN γ signature, and CD8+ T-cell abundance for high (yellow) and low (grey) purity tumors. Variables are presented in the order of AUC in high purity tumors and stars denote the significant empirical p-values.

These results are now described in the text (page 5) and Supplementary notes. Figure R1, Figure R2 and Figure R3 are included also in the revised manuscript as Figure S3, Figure S5 and Figure 5 respectively. Table R1 is also included in the revised manuscript as Table 1.

2. Focus on HLA peptidome part

Peptide sets used in the manuscript like “differentially presented peptides” or “immunogenic peptides”, that are central for the provided argumentation, should be properly defined. In the current form, elucidation of those definitions was extremely cumbersome and only possible by working through the supplementary information.

Response: We thank the reviewer for this comment. In the revised manuscript we clarified the description of the different peptide sets used.

The differential HLA peptidome analysis apparently was performed without any technical or biological replication. The reproducibility of the assay and statistical significance of differential MS1 signal intensities thus cannot be analyzed. In the current manuscript the authors simply consider any peptide with MS signal FC-values deviating from 1 as differentially presented, which is inappropriate. “Peptides differentially presented by IP OE cells” were apparently defined as peptide with either missing identification in EV or with OE/EV fold-change above 1 while “peptides differentially presented by EV cells” are either missing identification in OE or have a fold change below 1. This central limitation also renders the entire downstream analysis on differential immunogenicity of modulated peptides non-informative.

As conventionally employed in label-free quantification MS, the authors should conduct at least three experimental replicates per cell line (independent cell culture, transfection and HLA peptide isolation). LC-MS data acquisition should comprise technical replicates and be block randomized to control for potential spray instabilities and other technical variability which might not be corrected by normalization. Volcano Plots may then be shown to visualize the effect size and the significance levels of modulation.

Quality metrics of the HLA isolations should be presented by stating peptide yields and purities, e.g. as percentage of predicted binders or as percentage of clustered peptide vs. outliers in Gibbs clustering. Gibbs clustering parameter need to allow as many clusters as there are HLA allotypes expressed by the sample. The trash cluster option should be enabled. Presenting only a single Gibbs cluster for a class I peptidome comprising 6 HLA allotypes is not sufficiently informative. Peptides without HLA assignment or source cluster may be excluded from the differential HLA peptidome analysis as they likely constitute unspecific contaminants (for instance LDTNADKQLS which is present in every cell-line).

Response: Per the reviewer comments, we repeated the experiments and followed the suggestions above, as follows: (1) We first conducted three experimental replicates per cell line and condition. For cells with overexpression and vector control, we infected cells independently three times to produce independent cell cultures and performed three different HLA peptide isolations. For cells treated or non-treated with IFN γ we also collected three different pellets for each condition that was performed separately for each replicate. To avoid differences due to sample handling we processed each group of samples (OE and EV, or treated and non-treated) together – samples were handled on the same day, on the same batch of beads and were kept at similar conditions and same time prior to analysis on the mass spectrometer. Samples were block randomized to control for potential technical variability.

Before analyzing the data, we first used both NetMHCpan and Gibbs clustering to match the peptides to the allele they can be presented on. We analyzed the peptides using NetMHCpan 4.0 and kept only peptides with rank <2%. To classify the clustered peptides into HLA alleles, we first identified for each allele its motif. To do this, we retrieved all HLA-I epitopes registered under this allele from the Immune epitope database² (www.iedb.org, as of July 2019). All peptides were annotated as positive in “MHC ligand assays” to the specific HLA-I allele and between 8-13 amino acids long. The GibbsCluster 2.0 server (www.cbs.dtu.dk/services/GibbsCluster) was used to align the peptides using the “MHC class I ligands of length 8-13” parameters, the number of clusters was set to one and the trash cluster option was disabled. Each set of peptides (patient and treatment/ overexpression) was also clustered using Gibbs clustering, with the “MHC class I ligands of length 8-13” parameters, number of clusters was set to six and the trash cluster option was enabled (Supplementary Fig. 9-11). Since the number of peptides per each allele is different, for alleles with a higher number of peptides (as the HLA-A alleles) the unbiased clustering resulted in more than one cluster for these alleles. In these cases, we added all clusters that matched the allele motif in the same category. The change in number of peptides to allele also resulted in clusters with mixed motifs that were similar. In these cases, we assigned the cluster to the allele that had the highest representation in the cluster and added a note which other alleles are mixed within. We noticed that most peptides that are not assigned to a specific cluster were also not predicted to bind any HLA allele by NetMHCpan and were mostly longer. Those are probably unspecific contaminants

that we excluded from all further analyses. Per the reviewer's suggestion, we also prepared quality metrics of the HLA isolations, that includes the peptide yields and purities (percentage of predicted binders by NetMHCpan and percentage of clustered peptide vs. outliers in Gibbs clustering) in Supplementary Table 5. In all samples >98% of the peptides were clustered using Gibbs clustering and 89.2-95% of the peptides were predicted to bind the patients' HLA alleles using NetMHCpan. Peptides were further selected for the differential presentation analysis if they were assigned with HLA allele by both the Gibbs clustering and NetMHCpan predictions.

For the differential presentation analysis, we used the Perseus computational platform version 1.6.6.0. Peptide intensities were Log₂ transformed and missing intensity values were imputed by drawing random numbers from a Gaussian distribution with a standard deviation of 20% in comparison to the standard deviation of the measured peptide abundances (as in ³). Volcano plots which show differentially presented peptides of the relative intensities of HLA peptides between cells with overexpression of immunoproteasome subunits and empty vectors were created, as well as cells that were treated and non-treated with IFN γ (Fig. 3). The x axis represents the Log₂ fold changes of the peptide intensities, and the y axis represent the significance levels calculated by two-sided unpaired t test with a FDR of 0.05 and S₀ of 1. The peptides that were found to be significantly differentially presented between the overexpressing cells and the empty vector control (or the treated vs. non-treated cells) were used for further analysis of peptides' immunogenicity assay. As the number of TAAs is relatively low we analyzed all TAAs and then focused on the significantly differentially presented ones. All peptide identification data, as well as statistical analyses are now described in Supplementary Tables 6-10.

The HLA typing of the cell lines should be provided, the MS raw data should be made publically available.

Response: HLA typing as well as patient data is now provided in Supplementary Table 16. HLA peptidomics data have been deposited to the ProteomeXchange Consortium via the PRIDE partner repository with the dataset identifier PXD015957.

Because of their high potential impact (Bassani-Sternberg et al. Nat. Commun. 2016), putative IDs from neo-antigen need to be validated by calculating the spectral correlation with their synthetic counterparts and by co-elution with stable isotopically labeled peptides (see Blatnik et al., Proteomics 2018 [10.1002/pmic.201700390]).

Response: Thanks. To validate the neo-antigens that we have identified we compared their spectra with their synthetic counterpart and also spiked the sample with stable isotopically labeled peptide to show that they co-elute, as now described in Supplementary Fig. 14-15.

The hydrophobicity scoring method is not described, the effect size appears to be minute.

Differences in peptidome hydrophobicity profiles (Fig. 2) may also arise from different degrees of adsorption to autosampler vials due to different sample standing time prior to LC-MS as described by the CPTAC consortium (Hoofnagle et al., Clin Chem. 2016). The method section therefore should accurately describe sample handling, replication and statistical analysis as outlined in the Minimal Information about an Immuno-Peptidomics Experiment (MIAIPE) guidelines (Lill et al., Proteomics 2018) and as required by the reporting summary by Nature Communications.

It should be discussed if ligands restricted to HLA-A*02 are expected to be more immunogenic than those presented on e.g. HLA-A*03, because they are more hydrophobic due to their anchor amino acid usage.

Response: We thank the reviewer for this comment. Following, we checked again the properties of the differentially presented peptides and we see that there is preference to specific alleles after the overexpression as well as after treatment with IFN γ (which is known to increase mainly HLA-B alleles). As these effects may confound the analysis, we decided to remove this analysis from the revised manuscript.

No reproduction is shown for the IFN γ -ELISAs. For all 3 cell lines prevalence and intensity of IFN γ release is highly similar, irrespective of differential peptide presentation. The conclusion drawn from these data is heavily influenced by few outliers. These experiments require reproduction and adequate statistical evaluation to substantiate the association of differential immunogenicity of peptides associated with IP-overexpression.

The method section should accurately describe whether ELISAs were performed in parallel on the same plate for the different peptide populations. In addition, based on the provided summaries it can be deduced that “immunogenic peptides” were defined as peptides with IFN γ > 0 pg/ml which does not reflect the background signals observable in such assays.

Response: In the revised manuscript we added a more detailed description of the method. For each B-cell on which we loaded peptides, we also made sure to use a control of B-cell with DMSO (the vehicle in which the peptides were solubilized), which we treated in the same way as the B-cells that were loaded with peptides. The IFN γ concentration that was measured after co-culturing the control with the TILs was reduced from the values that were received with the peptides, and those numbers are the ones described in the figures (after reducing the background of B-cells + DMSO + TILs). Peptides with values >0 after removing the background signal are referred to as “immunogenic peptides”. All measurements were done in triplicates and we now added standard deviations to the pertaining plots. To compare between reads from different ELISA plates of the same experiment, we used a control peptide which was used to normalize the values of the different plates. We used plates from the same kit (and lot of production) to reduce such changes.

In the revised manuscript we carefully selected the differentially presented peptides (as described above), so now our reference to the difference in immunogenicity of the peptides is more accurate. We do expect to see immunogenic peptides also in the pool of peptides that are preferentially produced by the regular proteasome, this is in concordance with the literature (as we also referred to in the manuscript, in the introduction on page 3). Our goal was to show that the overexpression and the change in repertoire as a whole increases the immune response, which we agree was not clear in the original text. We now better clarify that the summed reactivity of the peptides produced preferentially by the immunoproteasome (the changed peptide repertoire) is greater, and not the reactivity of specific peptides. Furthermore, we also performed an additional experiment where we have compared the ability of the autologous TILs to kill the melanoma cells after IP overexpression compared to empty vector control. We do the same experiment also on cells treated with IFN γ and compare them to non-treated cells. These measurements were done in triplicates and we present the standard deviation in the graph. As can be seen in Fig. 2 in the main text, the ability of the TILs to kill the melanoma cells with overexpression of immunoproteasome or after IFN γ treatment is greater than their ability to kill the control cells and non-treated cells, respectively. These observations support our hypothesis that the overexpression of the immunoproteasome results in a superior immune response, and suggest that the antigens presented after these perturbations can induce a better immune response. We believe that this experiment is more adequate to show that the changes in the totality of the presented antigenic repertoire are indeed those that mediate the altered immune response observed.

Minor points:

- The claim of novelty (“We show here for the first time that the overexpression of these subunits is correlated with improved survival and better response to immune checkpoint inhibitors.”) should be toned down. Tripathi et al have previously shown similar results in NSCLC [10.1073/pnas.1521812113]. In general, the limitation of the findings to the utilized melanoma model should clearly be stated.

Response: We thank the reviewer for his comment. Throughout the manuscript we mention that our findings are in melanoma patients. We indeed missed it in the phrase you cite. We have now corrected the statement to be “We show here for the first time that the overexpression of these subunits is correlated with improved survival and better response to immune checkpoint inhibitors in melanoma” (page 3).

- Figure 1A: the legend for high and low IP expression is swapped

Response: We thank the reviewer for noticing this, it is now corrected.

- The statistical methods on the identification of PSMB8 and PSMB9 as a prognostic marker should be better explained:

- o If for z-transformation only the melanoma tumors were used, the finding that 15% are overexpressed at z-score>1 is an implicit consequence of the standard-normal-distribution of the Z-score (see 68–95–99.7-rule) and does not confirm over-expression.
- o If for z-transformation the mean and standard deviation was computed based on another reference data set (skin normals or other healthy tissues), this data set should be described and proper over-expression statistics (e.g. t-test) should be provided.

Response: Following the reviewer's comment, we also performed an additional analysis comparing to reference expression values from an independent cohort (using healthy skin samples in GTEX). We used the data with simultaneous normalization of TCGA and GTEX samples from UCSC Xena browser available at https://toil.xenahubs.net/download/TcgaTargetGtex_RSEM_Hugo_norm_count.gz. Our analysis shows that more than 76% of the samples show overexpression of PSMB8 and PSMB9 respectively, and we observe a significant overlap between PSMB8-overexpressed and PSMB9-overexpressed samples (hypergeometric $P < 4.9E-55$). These results are now reported in the main text on page 4.

- The Cox proportional hazards model was only applied to the dichotomized variable based on an arbitrary cutoff (top vs bottom 10%) which results in potential bias (See Eng et al. [10.18632/oncotarget.6121]). Thus, additional cutoffs should be displayed in the Kaplan-Meier plot and the Cox Hazard ratio and p-Value should be shown for the original continuous variable. It should also be shown for the mutational load.
- o The deltaAUC is not commonly used to quantify effect size in clinical research. Median OS should be shown instead.
- o Table S1: Instead of raw coefficients (beta), hazard ratios (HR) should be provided based on continuous expression values after z-transformation. Thus, HRs will be comparable between variables and be associated with change of 1 standard deviation rather than change of 1 unit of expression.

Response: As the reviewer pointed out, we originally performed the Kaplan-Meier (KM) analysis in a subset of TCGA melanoma samples comparing the patient survival of high IP samples (top 10 percentile) vs low IP samples (bottom 10 percentile). However, we want to note that the Cox regression in the original manuscript was done across all samples (N=356) for the expression values of PSMB8 and PSMB9.

To further confirm the robustness of our findings, following your request we now performed the KM analysis with different thresholds (25, 33, 50 percentile as thresholds), and the resulting KM curves are presented in Figure 1A,B and Figure S1 in the main text (and shown below in Figure R1E,F and Figure R5). The effect size is now quantified with the differences in median survival time of respective patient groups. As the reviewer noted, we originally reported the beta values (coefficients of the Cox regression) in previous Table S1, however, we note that each variable in the original Table S1 was already z-transformed so that the corresponding hazard ratios were comparable. We now explicitly report hazard ratios in the new Table S3.

A

B

C

D

E

F

Figure R5. Kaplan Meier (KM) analysis of IP expression in TCGA melanoma patients with varying thresholds. The KM plots compares the survival of patients with high (blue) vs low (yellow) levels of (A,C,E) PSMB8 expression and (B,D,F) PSMB9 expression. High (low) levels of PSMB8/9 were determined based on top 25-percentile (bottom 25-percentile) for (A,B), top tertile (bottom tertile) for (C,D), and top 50-percentile (bottom 50-percentile) for (E,F). Logrank p-values are denoted in the figure, and the table on the bottom summarizes the patients at risk in different time points. The effect size was quantified by the difference in medial survival of each patient group, which are 4379, 4094, 3,738, 4,025, 3083, and 3604 from A to F, respectively.

- Hydrophobicity data: As stated above, experimental confounders need to be precluded and the utilized score should be described. The correlation of the score with the experimental RTs should be analyzed. The statistics on the comparison of avg. hydrophobicity should be revised: The hydrophobicity scores are not normally distributed. The relevance of the rather small differences is questionable.

Response: We thank the reviewer for this comment. As we wrote before, due to confounding effects of the different alleles on the results we decided to remove this analysis from the revised manuscript.

- The numbers of Exome-seq based mutations for the 3 melanoma cell lines should be

provided. Identified NeoAg should be displayed separately in Figure 3A to clearly display their relative contribution.

Response: We now added the mutations list for these cell lines in Supplementary Table 14-15. We now marked more clearly where the neo-antigens are found in our analysis in the volcano plot.

- Figure 4: Different scales are used for the intensity ratio. The coloring not symmetrical for up- and down modulation.

Response: Thank you for noticing this. In the revised manuscript we changed the figures for peptide reactivity according to the reviewer's suggestions, so these modifications are no longer relevant.

- Computing the arithmetic mean of PSMB8 and PSMB9 is questionable if just Spearman correlation as a non-parametric statistic is shown. A scatterplot of both measurements and linear modelling should be provided to justify this step.

Response: Thanks. We agree to the reviewer's comment that arithmetic mean of PSMB8 and PSMB9 needs to be justified. Thinking about it more, as we now realized it is better to quantify the abundance of total immunoproteasome (IP) mRNAs by the summed expression levels of PSMB8 and PSMB9 rather than their mean. This is now clarified in the main text page 4.

- Not all authors are listed in the contribution section.

Response: We thank the reviewer for noticing. It was corrected in the revised manuscript.

Reviewer #2: Immunoproteasome and biomarkers (Remarks to the Author):

The manuscript by Kalaora et al addresses one of the largest outstanding questions in oncology – factors behind the heterogeneous response to immune checkpoint inhibition. This work focuses on a relatively new target, the immunoproteasome, as a predictor of this response. This hypothesis makes sense, in that antigen presentation, especially of tumor related neoantigens, is central to interaction between tumor cells and cytotoxic T-lymphocytes. The paper uses a variety of approaches, including analysis of TCGA data, cell culture models, and proteomic analyses of MHC-eluted peptides, to support their central hypothesis.

The paper is well written and I believe should be accepted with minor revision. The conclusions are novel, and builds on previous work showing altered immune responses with immunoproteasome expression in melanoma as well as differential expression of the immunoproteasome in cancer. This is supported, mostly, by the data presented. It will

have a good impact on the field and potentially alter treatment and outcomes.

Issues

- The TCGA analyses would be stronger if, from the outset, the authors made clear the tumor purity. Additionally, while the authors quantitate TILs and inflammatory cells, the immunoproteasome is expressed at high levels in antigen presenting cells including dendritic cells. While I see that other infiltrating and stromal cells are quantitated, is it possible to estimate APCs in these tumor specimens? If the immunoproteasome is predominantly in APCs, this could drastically alter the conclusions drawn in the paper. These tumors likely have higher levels of IFN-gamma, and thus could have a highly altered tumor microenvironment as well.

Response: In our original manuscript, we controlled for tumor purity in the patient survival analysis, but as per the reviewer's suggestion, we now explicitly consider tumor purity from the outset. To rule out the possibility that IP overexpression is a mere consequence of IFN γ signaling due to high immune cell fraction, we checked the correlation between IP expression and the IFN γ -related genes (IFNG, JAK1/2, and STAT1/2) or T-cell related genes. Our analysis shows that this correlation is significantly reduced in high purity tumors compared to the one in low purity tumors (Figure R6). This observation suggests that IP expression in tumor cells is independent of IFN γ signaling. This is further supported by the negative correlation over all samples between the estimated abundance of macrophages+dendritic cells and IP expression as shown in Figure R7. The anti-correlation between IP expression and APC abundance remains significant even when tumor purity is controlled for by employing a partial correlation analysis (PSMB8: Spearman R=-0.38, P<6.6E-18, PSMB9: Spearman R=-0.43, P<7.09E-23), indicating that IP expression is not likely originating from APCs. Our analysis further shows that the prediction accuracy of IP subunits is the highest in high purity tumors (Figure R8), while it is not the case in low purity tumors (Figure R9).

These issues have now been clarified and addressed on page 5 in the main text and Supplementary notes. The pertaining figures have been added in Figure S4 and Figure S5.

Figure R6. Correlation between IP and IFN-gamma related gene expression is lost in high purity tumors. Y-axis shows the correlation between the IP subunit expression and the expression of (A) IFN γ -related genes and (B) T-cell related genes in high (yellow; top 10-percentile) vs low (grey; bottom 10-percentile) tumors.

Figure R7 Correlation between IP and estimated abundance of APCs in TCGA melanoma. Y-axis shows the estimated abundance of macrophages and dendritic cells and X-axis shows (A) PSMB8 and (B) PSMB9 expression in TCGA melanoma. Spearman correlation was -0.52 and -0.59 ($P < 2.2E-16$), respectively.

Figure R8. IP subunits predicts the success of immune checkpoint therapy in high purity tumors. The Receiver Operating Curves (ROC) show that (A) IP expression itself shows the highest prediction accuracy for anti-PD1 therapy followed by mutational load, PDL1 expression, IFN γ signature, and CD8+ T-cell abundance. (B) IP expression shows the highest prediction accuracy for anti-CTLA4 therapy when combined with mutational load, followed by CD8+ T-cell abundance, IFN γ signature, and mutational load. We used the samples with tumor purity greater than 0.45, and this threshold was used throughout the paper.

Figure R9. IP subunits predicts the success of immune checkpoint therapy in high vs low purity tumors. The AUC of ROC curves (y-axis) is presented for (A) anti-PD1 and (B) anti-CTLA4 therapy for different variables IP subunit expression, mutational load, and their combination, PDL1 expression, IFN γ signature, and CD8+ T-cell abundance for high (yellow) and low (grey) purity tumors. Variables are presented in the order of AUC in high purity tumors and stars denote the significant empirical p-values.

- In the TCGA data, is there any correlation with PSMB10 expression? PSMB10, as well as some members of the TAP, have roles in immunoproteasome assembly and function.

Response: Thanks. Following the reviewer’s comment, we checked the association of PSMB10 and TAP1/2 expression with patient survival or immune checkpoint response. For patient survival, just like IFN γ -related genes or T-cell related genes, we performed univariate and multivariate Cox analysis in TCGA melanoma cohort. In parallel to IFN γ or T-cell genes, PSMB10 and TAP1/2 show significant association with patient survival in the univariate model (Table R2, 2nd and 3rd columns), but the significance gets reduced when tumor purity is controlled for (Table R2, 4th and 5th columns). For multivariate model, where IP subunit expression was considered together, PSMB10 and TAP1/2 expression completely lose the significant association with *better* survival and shows the trend in the direction of *worse* survival (hazard ratio>1), while IP subunit expression still shows association with *better* survival (though not strictly significant in all three cases). Furthermore, PSMB10 is not significantly associated with better response to ICT (Wilcoxon P>0.21) and TAP1/2 expression is not significantly associated with better response to anti-PD1 therapy (Wilcoxon P>0.23 and P>0.16, respectively), while PSMB8 and PSMB9 shows the association in both anti-PD1 and anti-CTLA4 as reported. These observations led us to focus on PSMB8 and PSMB9.

	Univariate model	Univariate model + purity	Multivariate model

	Hazard ratio	Wald P	Hazard ratio	Wald P	Hazard ratio	Wald P	Hazard ratio (of IP)	Wald P (of IP)
PSMB10	0.7081	0.0012	0.7728	0.0635	1.0118	0.9548	0.6710	0.0789
TAP1	0.6842	0.0006	0.7430	0.0357	1.1057	0.7382	0.6151	0.1382
TAP2	0.7007	0.0029	0.7881	0.1005	1.1369	0.5634	0.6094	0.0380

Table R2. Cox analysis for PSMB10 and TAP1/2. The table shows the Cox hazard ratio and corresponding Wald P-values for PSMB10, TAP1 and TAP2 gene expression in TCGA melanoma cohort. The hazard ratio and associated Wald P-values of the three genes (1st column) are presented in 2nd and 3rd columns for univariate model and 4th and 5th columns for univariate model with tumor purity. For multivariate model, which takes the independent variables of PSMB10, TAP1/2 expression together with tumor purity and IP subunit expression, the hazard ratio and Wald p-values for PSMB10 and TAP1/2 are shown in 6th and 7th columns and those for IP subunit expression are shown in 8th and 9th columns.

- I would like to see confirmation of expression of the immunoproteasome subunits in the overexpression cell lines. It would be useful to also have confirmation of enzymatic activity, perhaps through a commercially available luminescence assay. This would help understand what percentage of proteasome function in these overexpression cells is due to constitutive proteasome and which is due to immunoproteasome.

Response: Per the reviewer's suggestion we updated the western blot analysis figures of the cells with overexpression and IFN γ treated cells in Supplementary Fig. 6. We also performed an enzymatic activity assay to ensure the function of the immunoproteasome in the overexpression cells (Supplementary Fig. 7). We used fluorescent peptides that can be cleaved by chymotrypsin-like activity of PSMB5 and PSMB8 (Suc-LLVY-AMC), and a substrate that specifically cleaved by PSMB9 (Suc-PAL-AMC), which are commonly used for these assays. We observed an increase in the relative fluorescence unit (RFU) in the cells with the overexpression compared to the empty control of all three cell lines, indicating that the overexpressed immunoproteasome subunits were incorporated into the proteasome complex and are active. The change in RFU is correlated to the level of change in the immunoproteasome expression (Supplementary Fig. 6). We see the greatest change in 12T, which had the lowest expression of the immunoproteasome subunits in the empty vector cells (EV), then A375, and last 108T (which had the highest endogenous immunoproteasome expression in the empty vector cells). This additional analysis is now briefly described in the main text on page 6, which refers the reader to the pertaining Supplementary Fig. 6-7.

- Is there any quantitation of MHC expression, both in TCGA tumor specimens and in the IFN-gamma treated cells? Other components of the antigen presenting machinery are known to be highly induced by IFN-gamma.

Response: Thanks for this comment. In the revised manuscript we added flow cytometry quantification of membrane HLA expression of the cells that were treated with IFN γ as well as the cells that overexpress the immunoproteasome

subunits. As expected, we see an increase in HLA levels after IFN γ treatment and no change after overexpression, as briefly mentioned on page 6 in the main text and in Supplementary Fig. 8 in more detail.

In addition, Figure R10 shows the correlation between the gene expression of HLA-A and IP subunit expression. Indeed, we observe a strong correlation between HLA-A and IP subunit expression in low purity tumors, but this trend is significantly reduced in high purity tumors. Similarly as other IFN γ related genes (mentioned above) are not correlated with IP expression in high purity tumors, this observation suggests that the impact of *IP subunit expression in cancer cells* reported in our manuscript is not mere consequence of IFN γ secreted in the tumor environment.

Figure R10. Correlation between IP subunit and HLA-A expression is reduced in high purity tumors. Y-axis shows the correlation between the expression of HLA-A and IP expression in different subset of tumors of high purity (yellow) and low purity (gray) with varying thresholds in the purity in percentage (X-axis).

- A sentence or two describing the nature of the TAAs and neo-antigens examined would be useful. Were these point mutations? Were they deletions or frameshifts? Was there differences in presentation based on the mutation type?

Response: We identified two different neo-antigens, both were derived from missense mutations. We now better describe the identified neo-antigens in the text (page 8), and marked them in red in the volcano plots (Fig. 3) so it would be easier to see their fold of change in intensity between overexpression and empty vector cells as well as IFN γ treated and non-treated cells. The TAAs described in the manuscript are known cancer and melanoma antigens that were described previously⁴⁻⁶.

- In figures 4 and 5, I would appreciate a better discussion of the statistical methods used to judge statistical significance of IFN-gamma treatment. By dotplot, these looks to be driven by one or two outliers in each of these experiments. For the vast majority, IFN-gamma seems to be at the limit of detection.

Response: We thank the reviewer for this comment. We now changed the experimental design in order to add repetitions to the HLA peptidomics assay, and

changed the analysis to add statistics for the selection of significantly differentially presented HLA peptides. We first conducted three experimental replicates per each cell line and condition. For cells with overexpression of the IP subunits or a vector control, we infected cells independently three times to produce independent cell cultures and performed three different HLA peptide isolations. For cells treated or non-treated with IFN γ we also collected three different pellets for each condition that was performed separately for each replicate. We then filtered out all peptides that didn't assign to an HLA allele by both the Gibbs clustering and NetMHCpan predictions.

For the differential presentation analysis, we used the Perseus computational platform version 1.6.6.0. Peptide intensities were Log₂ transformed and missing intensity values were imputed by drawing random numbers from a Gaussian distribution with a standard deviation of 20% in comparison to the standard deviation of the measured peptide abundances (as in ³). Volcano plots which show differentially presented peptides of the relative intensities of HLA peptides between cells with overexpression of immunoproteasome subunits and empty vectors were created, as well as cells that were treated and non-treated with IFN γ (Fig. 3). This figure replaced Figures 4 and 5 in the original manuscript. The x axis represents the Log₂ fold changes of the peptide intensities, and the y axis represent the significance levels calculated by two-sided unpaired t test with a FDR of 0.05 and S0 of 1. The peptides that were found to be significantly differentially presented between the overexpressing cells and the empty vector control (or the treated vs. non-treated cells) were used for further analysis of peptides' immunogenicity assay. As the number of TAAs is relatively low we assessed all TAAs and then focused on the significant ones. All peptide identification data, as well as statistical analyses are found in Supplementary Tables 6-10.

In the revised manuscript we carefully selected the differentially presented peptides (as described above), so now our reference to the difference in immunogenicity of the peptides is more accurate. We do expect to see immunogenic peptides also in the pool of peptides that are preferentially produced by the regular proteasome, this is in concordance with the literature (as we also referred to in the manuscript, in the introduction on page 3). Our goal was to show that the overexpression and the change in repertoire as a whole increases the immune response, which we agree was not clear in the original text. We now better clarify that the summed reactivity of the peptides produced preferentially by the immunoproteasome (the changed peptide repertoire) is greater, and not the reactivity of specific peptides. Furthermore, we also performed an additional experiment where we have compared the ability of the autologous TILs to kill the melanoma cells after IP overexpression compared to empty vector control. We do the same experiment also on cells treated with IFN γ and compare them to non-treated cells. These measurements were done in triplicates and we present the standard deviation in the graph. As can be seen in Fig. 2 in the main text, the ability of the TILs to kill the melanoma cells with overexpression of immunoproteasome or after IFN γ treatment is greater than their ability to kill the control cells and non-treated cells,

respectively. These observations support our hypothesis that the overexpression of the immunoproteasome results in a superior immune response, and suggest that the antigens presented after these perturbations can induce a better immune response. We believe that this experiment is more adequate to show that the changes in the totality of the presented antigenic repertoire are indeed those that mediate the altered immune response observed.

Minor issues

- A few typos. In the “Cells” section of the methods line 152 “obtain” -> “obtained”
- Figure 4/5: The blue/black color scheme makes it difficult to discern which antigens are more immunogenic.
- Figure 6: the green boxplot in figure A is truncated

Response: We thank the reviewer for noticing these errors, they have been corrected in the revised manuscript.

Reviewer #3: Melanoma Immunotherapy (Remarks to the Author):

In this manuscript, Kalaora and colleagues advance an attractive hypothesis that the immunoproteasome subunits, which are IFN/stress inducible and alter the antigen presentation machinery, are differentially expressed in tumor cells and may contribute to shaping the anti-tumor immune response via immunoproteasome-specific neo-antigens.

They present the following pieces of data to support their hypothesis:

1. TCGA analysis of melanoma patients analyses by RNAseq for immunoproteasome expression - the top 10% vs bottom 10% show a difference in survival (which also corresponds to difference in "CYT" score or T cell infiltrate pattern).
- 2 MS and in vitro T cell stimulation assays on 3 melanoma cell lines (with matched blood) to show enhanced "immunogenicity" of immunoproteasome expressing tumors (which generate unique antigens).
3. Analysis of datasets of pts with melanoma treated with either CTLA-4 or PD-1 blockade showing that immunoproteasome expression (but not proteasome expression) is associated with better outcomes.

The main issue that the authors have not addressed is how important the IP expression is for influencing clinical outcomes - i.e. is increased elevation in IP levels simply a reflection of a more inflamed tumor environment that has already been correlated with better outcomes (IP is IFN induced). In that vein, it would be helpful to see a broader analysis of IFN-g responsive genes and a comparison to existing gene-signatures that have already been linked to clinical outcome in similar populations. In this dataset, are there "exceptions" that prove the rule - i.e. a tumor where IFN-g induced genes are upregulated, but IP subunits are lost or are not upregulated and the patient is resistant to checkpoint blockade ?

Response: Thanks for the insightful comment. As the reviewer noted in the summary paragraph, we “present a study on the association of immunoproteasome (IP) subunit expression as a proposed prognostic and predictive marker for melanoma patient survival and response to immune checkpoint inhibitor therapy.” Following the reviewer’s comment, we performed a systematic analysis comparing the performance of *IP subunit expression* to that of *T-cell infiltration* or *IFN γ signature* in predicting patient survival and the response to immune checkpoint therapy.

To this end we first extended our analysis on TCGA melanoma cohort from 356 samples in the original manuscript to 472 samples in the revised manuscript, which recently became available through the updated Genomic Data Common (GDC) TCGA data portal (<https://portal.gdc.cancer.gov>). Our patient survival analysis in this extended data shows immunoproteasome expression is associated with patients’ prognosis independent of IFN γ or T-cell infiltration.

- (1) As the reviewer pointed, we indeed observed IFN γ expression, IFN γ signature, T-cell related genes (e.g. CD8A) and computationally estimated CD8+ T-cell infiltration¹ show significant association with patient survival (Figure R11A,B,C,D). However, notably, this signal becomes insignificant when tumor purity is controlled for in the Cox model (Table R3).
- (2) In contrast, in the case of immunoproteasome, Both for PSMB8 and PSMB9, their expression is associated with better prognosis (Figure R11 E,F) and this association remains significant even when the tumor purity is controlled for (Table R3). Taken together with (1), these results indicate that IP expression significantly contributes to the survival independent of tumor purity, while IFN γ signature, IFNG expression, CD8A expression or CD8+ T-cell infiltration does not.
- (3) Finally, when we construct multivariate Cox model combining IP expression (summed expression of PSMB8/9) together with IFN γ signature and T-cell infiltration, we see significant association of IP expression with patient survival, but not for IFN γ or T-cell infiltration. This suggests that IP subunits are significantly associated with patient survival and this association is likely to be independent of IFN γ or T-cell infiltration (Table R3).

A

B

Number at risk

IFNG signature	91	23	12	6	4	2	1	1	0
Low	91	23	12	6	4	2	1	1	0
High	110	56	29	18	8	2	2	1	0

Time in days

Number at risk

IFNG expression	90	25	13	5	2	2	1	1	0
Low	90	25	13	5	2	2	1	1	0
High	106	53	28	19	9	2	2	1	0

Time in days

Number at risk

CD8+ T-cell	94	32	19	10	7	5	3	1	0
Low	94	32	19	10	7	5	3	1	0
High	111	46	24	15	6	0	0	0	0

Time in days

Number at risk

CD8A expression	97	27	15	8	5	4	2	1	0
Low	97	27	15	8	5	4	2	1	0
High	109	56	27	16	8	1	1	1	0

Time in days

Number at risk

PSMB8 expression	92	27	14	7	3	2	1	1
Low	92	27	14	7	3	2	1	1
High	115	39	19	13	5	0	0	0

Time in days

Number at risk

PSMB9 expression	93	25	12	6	3	2	1
Low	93	25	12	6	3	2	1
High	108	44	24	16	8	1	1

Time in days

Figure R11. Comparative Kaplan Meier (KM) analysis of IFN γ , T-cell infiltration, and IP expression in TCGA melanoma patients. The KM plots compare the survival of patients with high (top tertile, blue) vs low (bottom tertile, yellow) levels of (A) IFN γ signature, (B) IFNG expression, (C) estimated CD8+ T-cell abundance, (D) CD8A expression, (E) PSMB8 expression, and (F) PSMB9 expression. For each variable, the survival of patients with top tertile vs bottom tertile was compared using logrank test, and the p-values are denoted in the figure. The table on the bottom summarizes the patients at risk in different time points, and the median survival differences are 4025, 4025, 3115, 3738, 3738, and 4025, respectively.

	Univariate model		Univariate model + purity		Multivariate model	
	Hazard ratio	Wald P	Hazard ratio	Wald P	Hazard ratio	Wald P
IFN γ signature	0.709	0.000964	0.763	0.094754	1.042	0.861773
IFNG expression	0.691	0.001131	0.745	0.087037	1.116	0.679433
CD8+T-cell	0.766	0.015118	0.877	0.319634	1.097	0.576941
CD8A	0.694	0.000629	0.706	0.063550	0.886	0.628285
CD4	0.682	0.000536	0.678	0.078271	0.721	0.122652
CD3G	0.695	0.000953	0.695	0.114972	0.871	0.627104
Mutational load	0.796	0.074994	0.806	0.089139	0.842	0.194167
PSMB8	0.683	0.001128	0.753	0.036950	-	-
PSMB9	0.649	8.59E-05	0.651	0.009516	-	-
PSMB8/9	0.646	0.000138	0.677	0.011117	0.646	0.045741

Table R3. Comparative Cox regression analysis of IFN γ , T-cell infiltration, and IP expression in TCGA melanoma patients. The table shows the hazard ratio and corresponding Wald p-values of Cox univariate and multivariate models with IFN γ signature, IFNG expression, computationally estimated CD8+ T-cell infiltration, T-cell related gene expression (CD8A, CD4, and CD3G), PSMB8, and PSMB9 expression, and the summed expression of PSMB8/9 as independent variables for explaining patient survival.

To further strengthen our point, we checked the correlation between IP expression and the expression of IFN γ -related (Figure R12A) or T-cell related genes (Figure R12B) in high vs low purity tumors. We observe that the correlation between IP subunits and IFN γ genes or T-cell related genes is significantly reduced in high purity tumors compared to that in low purity tumors (Figure R12). This suggests that the IP subunit expression *in cancer cells* are not mere consequence of IFN γ release or T-cell infiltration, but rather independent of IFN γ release or T-cell infiltration.

Figure R12. Correlation between IP and IFN γ related gene expression is lost in high purity tumors. Y-axis shows the correlation between the IP subunit expression and the expression of (A) IFN γ -related genes and (B) T-cell related genes in high (yellow; top 10-percentile) vs low (grey; bottom 10-percentile) tumors.

We next compared the prediction accuracy for response to immune checkpoint therapies of (i) immunoproteasome expression to that of (ii) IFN γ signature, and (iii) T-cell infiltration. Our analysis shows that the prediction accuracy of IP subunits is the highest in high purity tumors (Figure R13), while it is not the case in low purity tumors (Figure R14). In combination with mutational load, IP subunit expression provides the highest prediction accuracy in both datasets of anti-PD1 and anti-CTLA4. This supports the notion that the IP subunit expression *in cancer cells* is predictive of immune checkpoint therapies response.

Figure R13. IP subunits predicts the success of immune checkpoint therapy in high purity tumors. The Receiver Operating Curves (ROC) show that (A) IP expression itself shows the highest prediction accuracy for anti-PD1 therapy followed by mutational load, PDL1 expression, IFN γ signature, and CD8+ T-cell abundance. (B) IP expression shows the highest prediction accuracy for anti-CTLA4 therapy when combined with mutational load, followed by CD8+ T-cell abundance, IFN γ signature, and mutational load. We used

the samples with tumor purity greater than 0.45, and this threshold was used throughout the paper.

figure R14. IP subunits predicts the success of immune checkpoint therapy in high vs low purity tumors. The AUC of ROC curves (y-axis) is presented for (A) anti-PD1 and (B) anti-CTLA4 therapy for different variables IP subunit expression, mutational load, and their combination, PDL1 expression, IFN γ signature, and CD8+ T-cell abundance for high (yellow) and low (grey) purity tumors. Variables are presented in the order of AUC in high purity tumors and stars denote the significant empirical p-values.

These results are now described in the text (page 5) and Supplementary notes. Figure R11, Figure R12 and Figure R13 are included also in the revised manuscript as Figure S3, Figure S5 and Figure 5 respectively. Table R3 is also included in the revised manuscript as Table 1.

Following the reviewer’s suggestion, we further checked if there exist exceptions where *IFN-g genes are up-regulated, but IP subunits are lost or are not upregulated and the patient is resistant to checkpoint blockade*. Since copy number data is not available for Van Allen et al. and Hugo et al. datasets, we first checked this potential association in TCGA cohort. Figure R15A shows the IP overexpressed (top 5-percentile) and underexpressed (bottom 5-percentile) samples (as evident from PSMB8/9 expression), and the corresponding copy number profiles of IFN γ related genes. As evident, there are cases where IP expression is high while IFN γ genes are lost (eg. TCGA-D3-A51K) and others where IP expression is low while IFN genes are amplified (eg. TCGA-D3-A1Q1). Figure R15B shows the samples with IFN γ genes are high (top 5-percentile) vs low (bottom 5-percentile), and the corresponding copy number profiles of IP genes. There are cases where IFN γ genes’ expression is low but IP genes are amplified. After observing the presence of these unexpected cases in TCGA melanoma cohort, we analyzed the RNAseq data in Van Allen et al. and Hugo et al. cohorts. In the Van Allen cohort (Figure R15C), there exist exceptions where IFN γ expression is high but IP expression is low, and the patient is resistant to anti-CTLA4 therapy (eg. Pat118). In Hugo cohort (Figure R15D), there also exist cases where IFN γ expression is high but IP expression is low, and the patient is resistant to anti-PD1 therapy (eg. Pt1), in the lines suggested by the referee.

A

Figure R15. IP and IFN γ related genes do not always correlate in TCGA and immune checkpoint therapy cohorts of melanoma. X-axis denotes samples and Y-axis denotes expression or copy number of PSMB8, PSMB9, IFN γ related genes or response annotation. (A) TCGA melanoma samples with high (top 5-percentile) and low (bottom 5-percentile) of IP expression, where the samples with low IP expression have amplification in IFN γ genes and vice versa. (B) TCGA melanoma samples with high (top

5-percentile) and low (bottom 5-percentile) of IFN γ signature, where the samples with low IFN γ signature have amplification in IP genes. (C) Van Allen et al. cohort treated with anti-CTLA4, where some of the nonresponders show high expression of IFN γ genes but low expression of PSMB8/9. (D) Hugo et al. cohort treated with anti-PD1, where some of the nonresponders show high expression of IFN γ genes but low expression of PSMB8/9.

Even with this complete analysis, challenging to causally link IP expression to clinical outcome using retrospective analysis of banked samples, and the authors should take care in describing the clinical data as exploratory and hypothesis generating.

Response: Thanks for the comment. By analyzing the extended TCGA data, our results point to a strong association between the expression of the IP subunit and patient survival, which remains significant after controlling for IFN γ expression and T-cell abundance. These results motivate and complement our further experimental and computational analyses of checkpoint response data but obviously, do not imply causality. We now make this point even more explicitly in the Discussion.

References

1. Newman, A.M. *et al.* Robust enumeration of cell subsets from tissue expression profiles. *Nat Methods* **12**, 453-7 (2015).
2. Vita, R. *et al.* The Immune Epitope Database (IEDB): 2018 update. *Nucleic Acids Res* **47**, D339-D343 (2019).
3. Chong, C. *et al.* High-throughput and Sensitive Immunopeptidomics Platform Reveals Profound Interferongamma-Mediated Remodeling of the Human Leukocyte Antigen (HLA) Ligandome. *Mol Cell Proteomics* **17**, 533-548 (2018).
4. Almeida, L.G. *et al.* CTdatabase: a knowledge-base of high-throughput and curated data on cancer-testis antigens. *Nucleic Acids Res* **37**, D816-9 (2009).
5. Vigneron, N., Stroobant, V., Van den Eynde, B.J. & van der Bruggen, P. Database of T cell-defined human tumor antigens: the 2013 update. *Cancer Immun* **13**, 15 (2013).
6. Andersen, R.S. *et al.* Dissection of T-cell antigen specificity in human melanoma. *Cancer Res* **72**, 1642-50 (2012).

Reviewers' Comments:

Reviewer #1:

Remarks to the Author:

The authors have addressed most of our initial points and have substantially strengthened the data basis of the manuscript. Upon addressing the following points, the manuscript may be considered for publication in Nature Communications.

The inclusion of quality metrics and experimental replicates in the HLA peptidome analysis together with adequate statistical analysis now enable the identification of differential HLA ligand presentation upon immunoproteasome (IP) overexpression or IFN γ treatment. However, the immunogenicity data still do not support one central conclusion that the IP / IFN-induced "change in [peptide] repertoire is responsible for the improved immune response in the tumor".

- Using the summed IFN γ release of OE vs EV associated peptides to compare the strength of the immune response may only be applicable if identical numbers of peptides are screened in both sets. The method also is highly prone to be influenced by outliers. Summary statistics such as the frequency of immune recognition and the median IFN γ release may be better suited to describe the data.

- The notation "differentially presented peptide" is still not clearly defined. If this set comprises all peptides except the "significantly differentially presented peptides", the word "differential" should be dropped from the notation as any difference of their abundance can be ascribed to biological/experimental/technical variation.

In our opinion the conclusion "change in [peptide] repertoire is responsible for the improved immune response in the tumor" should be toned down, in particular in the abstract.

Minor Points

- The comparison of predictive performance of IP expression compared to known correlates of ICT response is hampered by a low number of data points (n=26) rendering differences in AUC and absolute AUC values inaccurate. The exploratory character of this analysis should be more explicitly stated.

- ♣ There appears to be a problem with the color code in Figure 5B. The grey line is not annotated in the legend, the CD8+ T cells are missing in the plot

- ♣ The AUC values plotted for high purity tumors in Figure R4 of your response letter do not match with the AUCs reported in Figure R3 (Figure 5 in the paper).

- Validation of NeoAg IDs:

- o MED15 appears to show a clear spectral mismatch, however coelution was observed. Was a quality control run of the peptide performed using target-negative matrix and the same spike concentration to exclude contamination with unlabeled peptides. The MS2 fragment ion traces should be shown individually to enable the comparison of the fragmentation pattern of the co-eluting signal.

Reviewer #2:

Remarks to the Author:

The authors have extensively revised their manuscript on the role of the immunoproteasome in melanoma. While a study of this nature cannot conclusively show that increase in immunoproteasome activity is responsible for better prognosis in melanoma, they get about as near to this as possible, especially with the addition of new experiments and analyses. This, to my mind, addresses the major deficiency of the original manuscript. The authors specifically now include an IFN-gamma signature, tumor cell purity, and a CD8 signature as confounders, and still find a significant association between immunoproteasome expression and survival in melanoma.

An important finding is that, even with controlling for confounders, immunoproteasome expression

correlates better with response to immune checkpoint blockade inhibition than mutational burden or T-cell infiltration. The authors have conducted several new experiments, including more robust HLA and neoantigen proteomics, with a more advanced statistical analyses of the results. They have also expanded on in vitro work, including demonstrating functional activity of overexpressed proteasomal subunits and a more in-depth analyses of the immunogenicity of the putative neoantigens. This is a substantial revision and fully addresses my initial criticism.

Reviewer #3:

Remarks to the Author:

The authors have made significant revisions and additional analyses to address the concerns for this reviewer. No further comments and the manuscript is acceptable for publication.

Reviewers' comments:

Reviewer #1 (Remarks to the Author):

The authors have addressed most of our initial points and have substantially strengthened the data basis of the manuscript. Upon addressing the following points, the manuscript may be considered for publication in Nature Communications.

The inclusion of quality metrics and experimental replicates in the HLA peptidome analysis together with adequate statistical analysis now enable the identification of differential HLA ligand presentation upon immunoproteasome (IP) overexpression or IFN γ treatment. However, the immunogenicity data still do not support one central conclusion that the IP / IFN-induced “change in [peptide] repertoire is responsible for the improved immune response in the tumor”.

- Using the summed IFN γ release of OE vs EV associated peptides to compare the strength of the immune response may only be applicable if identical numbers of peptides are screened in both sets. The method also is highly prone to be influenced by outliers. Summary statistics such as the frequency of immune recognition and the median IFN γ release may be better suited to describe the data.

Response: We thank the reviewer for this comment. The working hypothesis underlying our study is that increased immunoproteasome activity results in a complete change in peptide repertoire, which we wished to comprehensively characterize and study in how it effects the immune response. To this end we set to evaluate **the complete repertoire of the peptides presented by melanoma cells** when the immunoproteasome is active using HLA peptidomics. We show that this repertoire is significantly altered in cells in which the immunoproteasome is active, leading to better killing of these cells by autologous TILs. This experiment takes into account the factors that affect the immune recognition – the level of presentation of the peptides, their binding, the abundance of T-cells and their activation state.

Therefore, we have not performed our reactivity experiments with a specific set of peptides and have not used the same number of peptides in each condition. For example, In Fig. 3, we show that 12T and 108T cells that overexpress immunoproteasome subunits now present more TAAs and neo-antigens (new peptides, or peptides with a higher intensity). After treatment with IFN γ , we see more TAAs in 12T and less TAAs in 108T. In all cases, whether the number of TAAs increased or decreased, we observed a higher **total reactivity** of the TILs toward all the TAAs, when tested separately, and against the cells that present those antigens. Indeed, in some cases the total reactivity was higher because there were more peptides in the group with immunoproteasome induction (OE or IFN γ), and/ or because a few specific peptides were presented more or uniquely after immunoproteasome induction. These TAAs or neoantigen “outliers” are the antigens that significantly improve the immune response.

Following the reviewer’s comment, we have now added a brief discussion of this issue in the Discussion section, noting the choice we have made here and explaining its motivation.

- The notation “differentially presented peptide” is still not clearly defined. If this set comprises all peptides except the “significantly differentially presented peptides”, the word “differential” should be dropped from the notation as any difference of their abundance can be ascribed to biological/experimental/technical variation.

In our opinion the conclusion “change in [peptide] repertoire is responsible for the improved immune response in the tumor” should be toned down, in particular in the abstract.

Response: We thank the reviewer for pointing this out. The term “differentially presented peptide” refers to all peptides that have an OE/EV intensity fold change greater or smaller than one. To clarify this, we changed the term from “differentially presented” to “all peptides”, and kept the term “significantly differentially presented” for the peptides that passed the statistical test.

Per the reviewer’s request, we have further changed the sentence in the Discussion from: “change in [peptide] repertoire is responsible for the improved immune response in the tumor” To: “Notably, however, our data suggest that enhanced IP activity in tumors plays an important role in the improved immune response”.

Minor Points

- The comparison of predictive performance of IP expression compared to known correlates of ICT response is hampered by a low number of data points (n=26) rendering differences in AUC and absolute AUC values inaccurate. The exploratory character of this analysis should be more explicitly stated.

Response: We thanks the reviewer for his comment. We understand the reviewer’s concern that the low sample size in our ICT analysis may call for further study in the future with extended data points. This point is now explicitly discussed as follows on page 15:

“Notably, the checkpoint datasets that are publicly available are still of modest size, calling for further studies in extended cohorts. Remarkably, however, the combination of IP expression with tumor mutational load predicts anti-CTLA4 response with almost 0.9 AUC and anti-PD1 response with an AUC of almost 0.8.”

§ There appears to be a problem with the color code in Figure 5B. The grey line is not annotated in the legend, the CD8+ T cells are missing in the plot

Response: Thanks for pointing this out. Indeed the grey line actually denotes CD8+ T cells. We have corrected this problem in the new Figure 5B (and the same for Figure S18 A).

§ The AUC values plotted for high purity tumors in Figure R4 of your response letter do not match with the AUCs reported in Figure R3 (Figure 5 in the paper).

Response: Thanks for pointing this out. Indeed, there were incorrect AUC values presented in Figure R4A (anti-PD1 dataset) originally, for which we sincerely apologize. The corrected figure is presented below. As shown before, the prediction accuracy of IP subunit expression and its combination with mutational load is significantly better in high purity tumors compared to low purity tumors.

Figure R4. IP subunits predicts the success of immune checkpoint therapy in high vs low purity tumors. The AUC of ROC curves (y-axis) is presented for (A) anti-PD1 and (B) anti-CTLA4 therapy for different variables IP subunit expression, mutational load, and their combination, PDL1 expression, IFN-gamma signature, and CD8+ T-cell abundance for high (>45%, yellow) and low (<45%, grey) purity tumors. The stars denote the significant empirical p-values.

- Validation of NeoAg IDs:

- o MED15 appears to show a clear spectral mismatch, however coelution was observed. Was a quality control run of the peptide performed using target-negative matrix and the same spike concentration to exclude contamination with unlabeled peptides. The MS2 fragment ion traces should be shown individually to enable the comparison of the fragmentation pattern of the co-eluting signal.

Response: Per the reviewer request, we now include in Supplementary Figure 15 a quality control run of the synthetic peptides alone to exclude contamination with unlabeled peptides. We also included the individual MS2 fragment ions to enable the comparison of the fragmentation pattern of the co-eluted peptides.

Reviewer #2 (Remarks to the Author):

The authors have extensively revised their manuscript on the role of the immunoproteasome in melanoma. While a study of this nature cannot conclusively show that increase in immunoproteasome activity is responsible for better prognosis in melanoma, they get about as near to this as possible, especially with the addition of new experiments and analyses. This, to my mind, addresses the major deficiency of the original manuscript. The authors specifically now include an IFN-gamma signature, tumor cell purity, and a CD8 signature as confounders, and still find a significant association between immunoproteasome expression and survival in melanoma.

An important finding is that, even with controlling for confounders, immunoproteasome expression correlates better with response to immune checkpoint blockade inhibition than mutational burden or T-cell infiltration. The authors have conducted several new

experiments, including more robust HLA and neoantigen proteomics, with a more advanced statistical analyses of the results. They have also expanded on in vitro work, including demonstrating functional activity of overexpressed proteasomal subunits and a more in-depth analyses of the immunogenicity of the putative neoantigens. This is a substantial revision and fully addresses my initial criticism.

Response: We deeply thank the reviewer for their appreciation of our manuscript.

Reviewer #3 (Remarks to the Author):

The authors have made significant revisions and additional analyses to address the concerns for this reviewer. No further comments and the manuscript is acceptable for publication.

Response: We deeply thank the reviewer for their appreciation of our manuscript.

Reviewers' Comments:

Reviewer #1:

Remarks to the Author:

The authors have addressed all our major points. The manuscript is acceptable for publication after addressing the following minor point:

NeoAg validation: There is an error in the newly added supplemental Fig15. Shown are still MS1 traces of the precursor and its isotopic peaks. MS2 data is not shown. Furthermore, the MS1 traces suggest that there is no co-elution of TPD52L2 with its labeled counterpart. The monoisotopic peak of the natural peptide shows shifted RTs. The dominant M+1 and M+2 peaks appear to belong to a different peptide species.

The authors should clarify this by showing the MS2 data. If the data is not supportive, the authors may still consider including this information as a note for the relevance of stringent neoAg validation in HLA peptidomics.

REVIEWERS' COMMENTS:

Reviewer #1 (Remarks to the Author):

The authors have addressed all our major points. The manuscript is acceptable for publication after addressing the following minor point:

NeoAg validation: There is an error in the newly added supplemental Fig15. Shown are still MS1 traces of the precursor and its isotopic peaks. MS2 data is not shown. Furthermore, the MS1 traces suggest that there is no co-elution of TPD52L2 with its labeled counterpart. The monoisotopic peak of the natural peptide shows shifted RTs. The dominant M+1 and M+2 peaks appear to belong to a different peptide species.

The authors should clarify this by showing the MS2 data. If the data is not supportive, the authors may still consider including this information as a note for the relevance of stringent neoAg validation in HLA peptidomics.

Response: We thank the reviewer for approving the publication of the manuscript and appreciate the insightful comments we received throughout the revision process. We have now corrected the figure and added the MS2 data of the peptides (Supplementary Figure 15).